# VISION-BASED MANIPULATORS NEED TO ALSO SEE FROM THEIR HANDS

**Kyle Hsu**[*], **Moo Jin Kim**[*], **Rafael Rafailov, Jiajun Wu, Chelsea Finn**
Stanford University
{kylehsu,moojink,rafailov,jiajunwu,cbfinn}@cs.stanford.edu

## ABSTRACT

We study how the choice of visual perspective affects learning and generalization in the context of physical manipulation from raw sensor observations. Compared with the more commonly used global third-person perspective, a hand-centric (eye-in-hand) perspective affords reduced observability, but we find that it consistently improves training efficiency and out-of-distribution generalization. These benefits hold across a variety of learning algorithms, experimental settings, and distribution shifts, and for both simulated and real robot apparatuses. However, this is only the case when hand-centric observability is sufficient; otherwise, including a third-person perspective is necessary for learning, but also harms out-of-distribution generalization. To mitigate this, we propose to regularize the third-person information stream via a variational information bottleneck. On six representative manipulation tasks with varying hand-centric observability adapted from the Meta-World benchmark, this results in a state-of-the-art reinforcement learning agent operating from both perspectives improving its out-of-distribution generalization on every task. While some practitioners have long put cameras in the hands of robots, our work systematically analyzes the benefits of doing so and provides simple and broadly applicable insights for improving end-to-end learned vision-based robotic manipulation.[1]

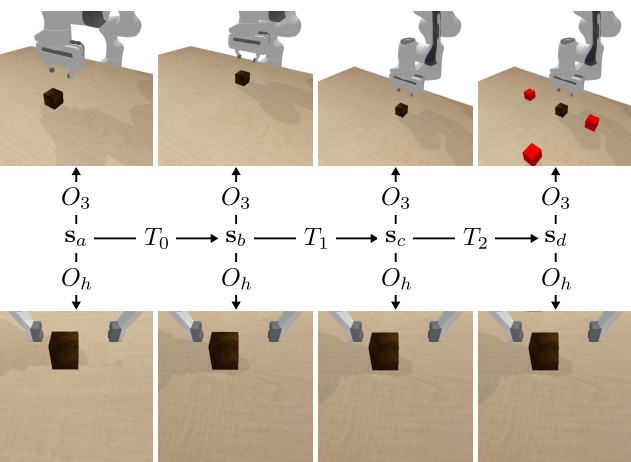

Figure 1: Illustration suggesting the role that visual perspective can play in facilitating the acquisition of symmetries with respect to certain transformations on the world state $\mathbf{s}$. $T_0$: planar translation of the end-effector and cube. $T_1$: vertical translation of the table surface, end-effector, and cube. $T_2$: addition of distractor objects. $O_3$: third-person perspective. $O_h$: hand-centric perspective.

## 1 INTRODUCTION

Physical manipulation is so fundamental a skill for natural agents that it has been described as a "Rosetta Stone for cognition" (Ritter & Haschke, 2015). How can we endow machines with similar

---

[*]Co-first authorship. Order determined by coin flip.
[1]Project website: https://sites.google.com/view/seeing-from-hands.

mastery over their physical environment? One promising avenue is to use a data-driven approach, in which the mapping from raw sensor observations of the environment (and other readily available signals, e.g. via proprioception) to actions is acquired inductively. Helpful inductive biases in modern machine learning techniques such as over-parameterized models and stochastic gradient descent have enabled surprising (and poorly understood) generalization capabilities in some applications (Neyshabur et al., 2014; Belkin et al., 2019; Zhang et al., 2021). Despite this, visuomotor policies learned end-to-end remain brittle relative to many common real-world distribution shifts: subtle changes in lighting, texture, and geometry that would not faze a human cause drastic performance drops (Julian et al., 2020).

While a wide variety of algorithms have been proposed to improve the learning and generalization of object manipulation skills, in this paper we instead consider the design of the agent's observation space, a facet of the learning pipeline that has been underexplored (Section 5). Indeed, in some applications of machine learning, e.g., image classification or text summarization, the disembodied nature of the task affords relatively little flexibility in this regard. Yet, even in these settings, simple data processing techniques such as normalization and data augmentation can have noticeable effects on learning and generalization (Perez & Wang, 2017). The role of data can only be more profound in an embodied setting: any sensors capable of being practically instrumented will only provide a partial observation of the underlying world state. While partial observability is typically regarded as a challenge that only exacerbates the difficulty of a learning problem (Kaelbling et al., 1998), we may also consider how partial observations can facilitate the acquisition of useful symmetries.

The natural world gives clear examples of this. For instance, because cutaneous touch is inherently restricted to sensing portions of the environment in direct contact with the agent, tactile sensing by construction exhibits invariances to many common transformations on the underlying world state; grasping an apple from the checkout counter (without looking at it) is largely the same as doing so from one's kitchen table. Due in part to the nascent state of tactile sensing hardware (Yuan et al., 2017) and simulation (Agarwal et al., 2020), in this work we investigate the above insight in vision, the ubiquitous sensory modality in robotic learning. In particular, we focus on the role of *perspective* as induced from the placement of cameras. To roughly imitate the locality of cutaneous touch, we consider the hand-centric (eye-in-hand) perspective arising from mounting a camera on a robotic manipulator's wrist. We also consider the more commonly used third-person perspective afforded by a fixed camera in the world frame.

The main contribution of this work is an empirical study of the role of visual perspective in learning and generalization in the context of physical manipulation. We first perform a head-to-head comparison between hand-centric and third-person perspectives in a grasping task that features three kinds of distribution shifts. We find that using the hand-centric perspective, with no other algorithmic modifications, reduces aggregate out-of-distribution failure rate by $92\%$, $99\%$, and $100\%$ (relative) in the imitation learning, reinforcement learning, and adversarial imitation learning settings in simulation, and by $45\%$ (relative) in the imitation learning setting on a real robot apparatus.

Despite their apparent superiority, hand-centric perspectives cannot be used alone for tasks in which their limited observability is a liability during training. To realize the benefits of hand-centric perspectives more generally, we propose using both hand-centric and third-person perspectives in conjunction for full observability while regularizing the latter with a variational information bottleneck (Alemi et al., 2016) to mitigate the latter's detrimental effects on out-of-distribution generalization. We instantiate this simple and broadly applicable principle in DrQ-v2 (Yarats et al., 2021), a state-of-the-art vision-based reinforcement learning algorithm, and find that it reduces the aggregate out-of-distribution failure rate compared to using both perspectives naively by $64\%$ (relative) across six representative manipulation tasks with varying levels of hand-centric observability adapted from the Meta-World benchmark (Yu et al., 2020).

## 2 PROBLEM SETUP

**Preliminaries: MDPs and POMDPs.** We frame the physical manipulation tasks considered in this work as discrete-time infinite-horizon Markov decision processes (MDPs). An MDP $\mathcal{M}$ is a 6-tuple $(\mathcal{S}, \mathcal{A}, P, R, \gamma, \mu)$, where $\mathcal{S}$ is a set of states, $\mathcal{A}$ is a set of actions, $P : \mathcal{S} \times \mathcal{A} \to \Pi(\mathcal{S})$ is a state-transition (or dynamics) function, $R : \mathcal{S} \times \mathcal{A} \to \mathbb{R}$ is a reward function, $\gamma \in (0, 1)$ is a discount factor, and $\mu \in \Pi(\mathcal{S})$ is an initial state distribution. An MDP whose state cannot be directly observed can be formalized as a partially observable MDP (POMDP), an 8-tuple $(\mathcal{S}, \mathcal{A}, P, R, \gamma, \mu, \Omega, O)$

that extends the underlying MDP with two ingredients: a set of observations $\Omega$ and an observation function $O : \mathcal{S} \times \mathcal{A} \rightarrow \Pi(\Omega)$. We consider only a restricted class of POMDPs in which the observation function is limited to be $O : \mathcal{S} \rightarrow \Omega$. To solve a POMDP, we optimize a policy $\pi : \Omega \rightarrow \Pi(\mathcal{A})$ to maximize the expected return $\boldsymbol{R}(\mathcal{M}, \pi \circ O) = \mathbb{E}_{\mu, P, \pi} \left[ \sum_{t=0}^{\infty} \gamma^t R(s_t, a_t) \right]$, where $\pi \circ O$ maps a state to an action distribution via composing the policy and observation function.

**Observation functions.** In this work, we denote the observation functions corresponding to the hand-centric and third-person visual perspectives as $O_h$ and $O_3$, respectively. We also consider proprioception, denoted as $O_p$. Often, multiple observation functions are used together; for example, we denote using both the hand-centric and proprioceptive observations as $O_{h+p}$.

**Invariances and generalization.** We say that a function $f : \mathcal{X} \times \mathcal{Y} \rightarrow \mathcal{Z}$ is invariant in domain subspace $\mathcal{X}$ to a transformation $T : \mathcal{X} \rightarrow \mathcal{X}$ iff $\forall x \in \mathcal{X}, y \in \mathcal{Y}. f(T(x), y) = f(x, y)$. We formalize the notion of generalization by saying that $\pi \circ O$ generalizes in $\mathcal{M}$ to a distribution shift caused by transformation $T$ iff $\boldsymbol{R}(\mathcal{M}, \pi \circ O)$ is invariant in $\mathcal{M}$ to $T$. We consider two kinds of generalization: in-distribution and out-of-distribution generalization, also referred to as interpolation and extrapolation. The latter corresponds to the agent generalizing in $\mathcal{M}$ to some specified transformation, and the former is a special case when the transformation is identity. In this work, we limit the scope of the transformations on $\mathcal{M}$ we consider to those acting on the initial state distribution $\mu$ through the state set $\mathcal{S}$. A few concrete examples of such transformations are illustrated in Figure 1.

## 3 HAND-CENTRIC VS. THIRD-PERSON PERSPECTIVES

The first hypothesis we investigate is that using the hand-centric perspective $O_h$ instead of the third-person perspective $O_3$ can significantly improve the learning and generalization of the agent $\pi \circ O$. In this section, we probe this hypothesis in settings where the hand-centric perspective gives sufficient observability of the scene (we consider when this does not hold in Section 4).

### 3.1 SIMULATED EXPERIMENTS

We first consider a visuomotor grasping task instantiated in the PyBullet physics engine (Coumans & Bai, 2016–2021). A simulated Franka Emika Panda manipulator is tasked with picking up a specific cube that initially rests on a table. The action space is 4-DoF, consisting of 3-DoF end-effector position control and 1-DoF gripper control. Observation functions include $O_h$ and $O_3$, which output $84 \times 84$ RGB images, and $O_p$, which outputs 3D end-effector position relative to the robot base, 1D gripper width, and a Boolean contact flag for each of two gripper "fingers".

We use three learning algorithms: imitation learning with dataset aggregation (DAgger) (Ross et al., 2011), reinforcement learning using data-regularized Q-functions (DrQ) (Kostrikov et al., 2020), and adversarial imitation learning using discriminator-actor-critic (DAC) (Kostrikov et al., 2018). We defer exposition on these algorithms to Appendix A.2. We run DAgger and DrQ on three experiment variants that each target a test-time distribution shift in the table height, distractor objects, and table texture. The distribution shifts are detailed and visualized in Appendix A.1. With DAC, we assess in-distribution generalization in the training environment and out-of-distribution generalization between demonstration (demo) collection and the training environment. Details on the model architectures and hyperparameters used can be found in Appendices A.3 and A.4. DAgger and DrQ results are reported in Figure 2 and aggregated in Table 1. DAC results are reported in Figure 3, with experiment variant descriptions in the caption.

For DAgger (left two columns of Figure 2), we find that the hand-centric perspective leads to clear improvements in out-of-distribution generalization (test) across all three experiment variants despite in-distribution generalization progress (train) being essentially identical between $\pi \circ O_{h+p}$ and $\pi \circ O_{3+p}$. The only exceptions to $\pi \circ O_{h+p}$ generalizing better are in some instances of the distractor objects variant. Here, seeing the red, green, and blue distractor objects during training was sufficient for both $\pi \circ O_{h+p}$ and $\pi \circ O_{3+p}$ to learn to ignore these object colors, even under distractor distribution shift. Generalization to white distractors was likely facilitated by the RGB representation of white as the "sum" of red, green, and blue.

For DrQ (right two columns of Figure 2), the differences between $\pi \circ O_{h+p}$ and $\pi \circ O_{3+p}$ extend into training time. In the table height variant, $\pi \circ O_{h+p}$ exhibits increased sample efficiency for training as well as similar out-of-distribution generalization benefits as seen for DAgger. For the distractor objects variant, $\pi \circ O_{h+p}$ converges before $\pi \circ O_{3+p}$ makes any significant progress on success

Table 1: Aggregate cube grasping out-of-distribution generalization performance across the three experiment variants computed from taking the three highest success rates achieved in each run of each DAgger or DrQ agent. The hand-centric perspective leads to the best aggregate success rate for both algorithms, with its 95% confidence intervals of the interquartile mean (IQM) not overlapping those of the third-person perspective.

| | | success rate (%) | | | |
|---|---|---|---|---|---|
| | | mean | median | IQM | 95% CI of IQM |
| DAgger | $\pi \circ O_{h+p}$ | **92.2** | **99.4** | **97.5** | $[\mathbf{97.0}, \mathbf{97.8}]$ |
| | $\pi \circ O_{3+p}$ | 64.4 | 61.1 | 69.3 | $[67.0, 71.4]$ |
| DrQ | $\pi \circ O_{h+p}$ | **94.7** | **100.0** | **99.3** | $[\mathbf{98.4}, \mathbf{99.9}]$ |
| | $\pi \circ O_{3+p}$ | 46.6 | 59.4 | 46.9 | $[44.3, 49.3]$ |

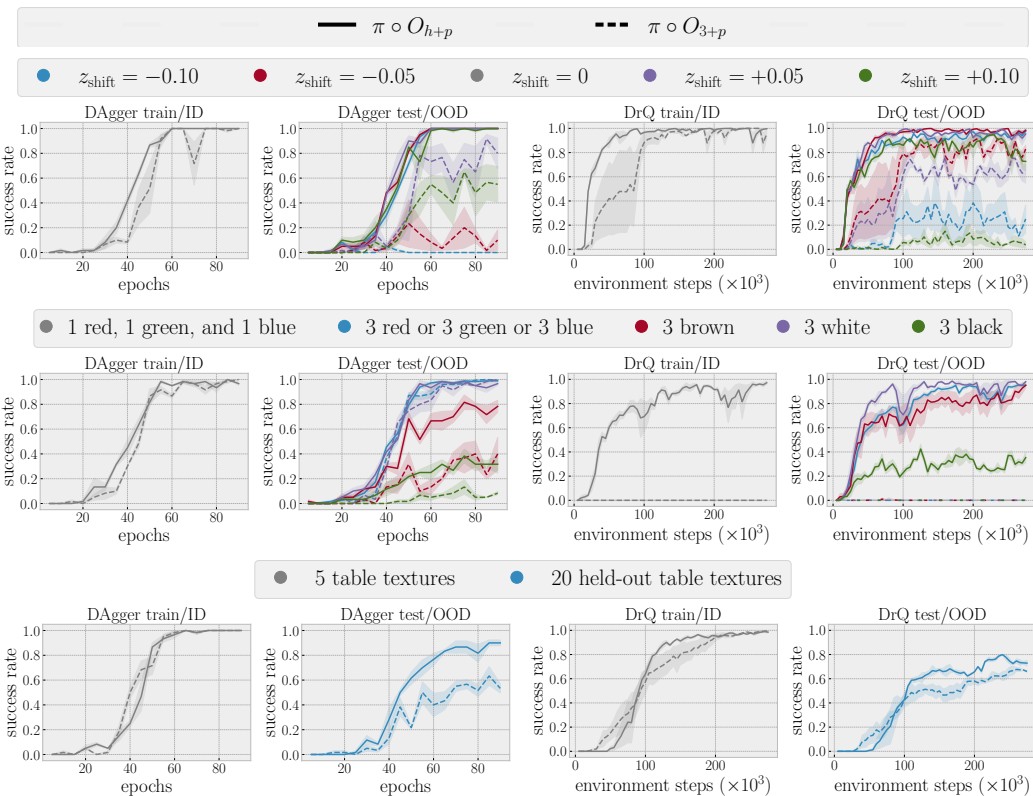

Figure 2: DAgger and DrQ results for cube grasping. The first, second, and third rows respectively contain results for the table height (shifted by $z_{\text{shift}}$), distractor objects, and table textures experiment variants. See Appendix A.1 for visualizations of the train and test distributions for each experiment variant. Compared to the third-person perspective (dashed lines), the hand-centric perspective (solid lines) leads to better out-of-distribution generalization performance across all three distribution shifts for both DAgger and DrQ. For DrQ, we also see appreciable improvements in sample efficiency when using the hand-centric perspective. Shaded regions indicate the standard error of the mean over three random seeds.

rate (though we did observe increasing returns). Since DrQ trained $\pi \circ O_{3+p}$ to convergence for the other variants within the same interaction budget, it follows that the presence of the distractors rendered the training task too hard for $\pi \circ O_{3+p}$, but not for $\pi \circ O_{h+p}$. In the table textures variant, the generalization improvement of $\pi \circ O_{h+p}$ over $\pi \circ O_{3+p}$ is less extreme. We attribute this to invariances to image-space transformations learned via the data augmentation built into DrQ. In Appendix A.5, an ablation in which this augmentation is removed further shows its importance.

For DAC, we find stark improvements in the generalization of $\pi \circ O_h$ over that of $\pi \circ O_3$. In the first DAC-specific experiment variant (left plot of Figure 3), $\pi \circ O_h$ fully generalizes in-distribution with as few as 5 demos, whereas $\pi \circ O_3$ achieves significantly lower success, even with 25 demos and much more online interaction. In the second variant (center plot of Figure 3), the distribution shift between demo collection and training barely affects $\pi \circ O_h$, but severely compromises the

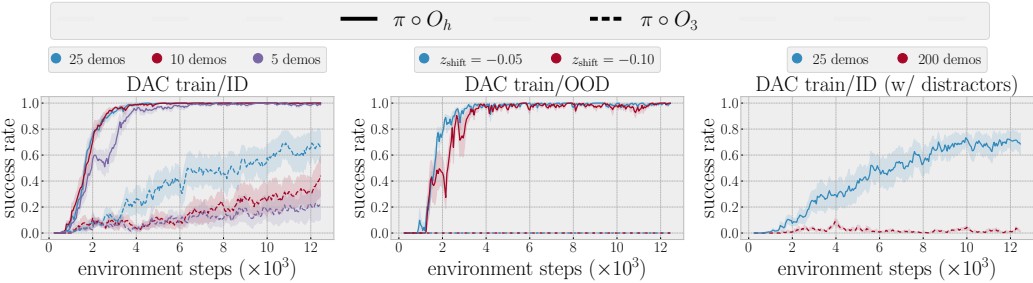

Figure 3: DAC results for cube grasping. Left: base variant (initial object and end-effector position randomization) with no distribution shift between demo collection and training. Center: base variant with table height shift between collection of 25 demos and training. Right: base variant plus three distractor objects with no distribution shift between demo collection and training. Across the three experiment variants, the hand-centric perspective enables the agent to generalize in- and out-of-distribution more efficiently and effectively. Shaded regions indicate the standard error of the mean over five random seeds.

training of $\pi \circ O_3$. In the third variant (right plot of Figure 3), despite the presence of distractor objects giving the discriminator strong predictive power in distinguishing between demos and agent behavior, $\pi \circ O_h$ still achieves a significant measure of in-distribution generalization, whereas $\pi \circ O_3$ makes little progress even with eight times the number of demos. We remark that, in the context of adversarial imitation learning, $\pi \circ O_h$ achieves its sample efficiency and robustness without any special requirements on the training data (Zolna et al., 2020) or modified training objectives (Xu & Denil, 2020).

## 3.2 REAL ROBOT EXPERIMENTS

We further investigate our hypothesis in a real-world analogue of the above environment: a Franka Emika Panda manipulator equipped with a parallel-jaw gripper is tasked with grasping a Scotch-Brite sponge amongst distractors (Figure 4). The action space consists of 3-DoF end-effector position control and 1-DoF gripper control. $O_h$ and $O_3$ output $100 \times 100$ RGB images, and $O_p$ outputs the 3D end-effector position relative to the robot base and the 1D gripper width. We train $\pi \circ O_{h+p}$ and $\pi \circ O_{3+p}$ via behavior cloning (BC) on 360 demonstrations collected via teleoperation, obtaining 85% success rate on the training distribution for both. Like above, we consider test-time distribution shifts in the table height, distractor objects, and table texture. Assessment of each distribution shift instance was done using 20 sampled environment initializations. Appendix B presents the setup in full detail as well as results stratified by distribution shift. Table 2 summarizes the results. Videos are available on our project website. These experiments indicate that the hand-centric perspective better facilitates out-of-distribution generalization for visuomotor manipulation not only in simulation, but also on a real robot.

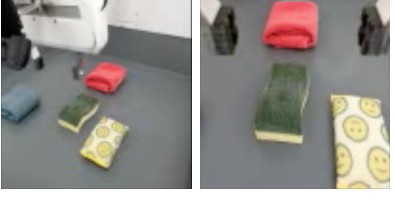

Figure 4: Sample observations from $O_3$ (left) and $O_h$ (right) in our real robot apparatus.

Table 2: Aggregate sponge grasping out-of-distribution generalization performance across the three sets of distribution shifts.

| | success rate (%) | | |
| --- | --- | --- | --- |
| | mean | median | IQM |
| $\pi \circ O_{h+p}$ | **52.0** | **52.5** | **53.3** |
| $\pi \circ O_{3+p}$ | 20.0 | 12.5 | 15.8 |

## 4 INTEGRATING HAND-CENTRIC AND THIRD-PERSON PERSPECTIVES

The previous experiments demonstrate how hand-centric perspectives can lead to clear improvements in learning and generalization over third-person perspectives. Unfortunately, this does not mean that the use of hand-centric perspectives is a panacea. The limited observability of hand-centric perspectives is a double-edged sword: depending on the environment and task, it can enable $\pi \circ O_h$ to establish useful invariances, or confuse $\pi \circ O_h$ by enforcing harmful ones. In this section, we focus on evaluating across tasks of varying hand-centric observability, including those in

which insufficient observability severely undermines $\pi \circ O_h$. How can we realize the benefits of hand-centric perspectives even in such scenarios?

## 4.1 Regularizing the Third-Person Information Stream

Insufficient observability arising from using $O_h$ alone necessitates the inclusion of $O_3$. While using both perspectives should effectively resolve the issue of insufficient observability and enable the agent to train, we know from Section 3 that the use of the third-person perspective can hamper out-of-distribution generalization by allowing the agent to "overfit" to particularities of the training distribution. To mitigate this, we propose to regularize the third-person perspective's representation. While multiple regularization techniques could conceivably be suitable to this end, we choose the variational information bottleneck (VIB) to use in our experiments due to its simplicity, theoretical justification, and empirical performance (Alemi et al., 2016).

For our subsequent experiments, we build on top of the state-of-the-art vision-based actor-critic reinforcement learning algorithm DrQ-v2 (Yarats et al., 2021) (see Appendix C.3 for a detailed description). When we use both hand-centric and third-person observations $\mathbf{o}_h$ and $\mathbf{o}_3$, we instantiate two separate image encoders $f_{\xi_h}$ and $f_{\xi_3}$. We denote the corresponding representations as $\mathbf{z}_h$ and $\mathbf{z}_3$. These are concatenated before being fed to the actor $\pi_\phi$ and critic networks $Q_{\theta_1}, Q_{\theta_2}$.

We apply a VIB to the third-person information stream to regularize the DrQ-v2 critic. This amounts to a variational approximation to maximizing the mutual information between the third-person observations and the critic's predictions of the temporal difference targets while minimizing the mutual information between the third-person observations and their representations. We implement this by replacing the deterministic third-person encoder $f_{\xi_3}$ with a stochastic encoder $p_{\xi_3}(\mathbf{z}_3|\mathbf{o}_3)$, specifying a prior $p(\mathbf{z}_3)$, and adding a weighted KL divergence term to the critic loss. The VIB-regularized DrQ-v2 critic objective is

$$\mathcal{L}(\xi_h, \xi_3, \theta_1, \theta_2) = \mathbb{E}_{\mathcal{D}, p_{\xi_3}}\left[\mathcal{L}_{\text{DrQ-v2 critic}}(\xi_h, \xi_3, \theta_1, \theta_2)\right] + \mathbb{E}_{\mathcal{D}}\left[\beta_3 D_{\text{KL}}(p_{\xi_3}(\mathbf{z}_3|\mathbf{o}_3) \parallel p(\mathbf{z}_3))\right], \quad (1)$$

where $\mathcal{D}$ is the replay buffer. We specify $p_{\xi_3}(\mathbf{z}_3|\mathbf{o}_3)$ as a diagonal Gaussian and $p(\mathbf{z}_3)$ as a standard Gaussian, which enables analytical computation of the KL divergence. We use the reparameterization trick to enable optimization of the first term via pathwise derivatives. We do not need to modify the actor objective as only gradients from the critic are used to update the encoder(s) in DrQ-v2. We remark that a (variational) information bottleneck can be applied to many imitation learning and reinforcement learning algorithms (Peng et al., 2018; Goyal et al., 2019; Igl et al., 2019; Kumar et al., 2021).

## 4.2 Meta-World Experimental setup

We evaluate the learning and generalization performance of seven DrQ-v2 agents: $\pi \circ O_{h+p}$ (hand-centric perspective), $\pi \circ O_{3+p}$ (third-person perspective), $\pi \circ O_{h+3+p}$ (both perspectives), $\pi \circ O_{h+3+p} + \text{VIB}(\mathbf{z}_3)$ (both perspectives with a VIB on the third-person information stream), and three ablation agents introduced later. We evaluate the agents on six tasks adapted from the Meta-World benchmark (Yu et al., 2020). We design the task set to exhibit three levels of hand-centric observability (high, moderate, and low) with two tasks per level. In each task, a simulated Sawyer robot manipulates objects resting on a table. The action space is 4-DoF, consisting of 3-DoF end-effector position control and 1-DoF gripper control. We do not use the original Meta-World observation space as it contains low-dimensional pose information about task-pertinent objects instead of images. Rather, we configure the observations so that $O_h$ and $O_3$ output $84 \times 84$ RGB images, and $O_p$ outputs 3D end-effector position and 1D gripper width. See Figure 5 for a visualization of each task through the lens of $O_h$ and $O_3$. Experiments in Appendix C.5 establish that proprioception alone is not sufficient to reliably solve any of the tasks. Experiments in Appendix C.6 consider variations of peg-insert-side that require an additional 1-DoF end-effector orientation control.

While the distribution shifts in the experiments of the previous section arise from transformations on the table height, distractor objects, and table textures, in this section we focus on distribution shifts arising from transformations on the initial object positions. All object positions have disjoint initial train and test distributions such that the latter's support "surrounds" that of the former (see Table 9 in Appendix C.2 for details).

Aside from adapting the DrQ-v2 algorithm to our setting as described above, we use the original DrQ-v2 model and hyperparameters with some minor exceptions (see Appendix C.7 for details).

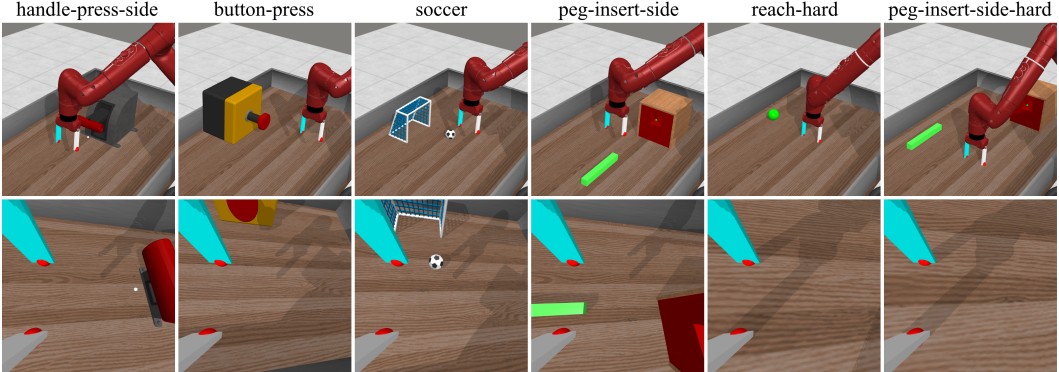

Figure 5: The Meta-World tasks used in the experiments in Section 4. The top row contains third-person observations $\mathbf{o}_3$, and the bottom row contains corresponding hand-centric observations $\mathbf{o}_h$. Initial object positions are randomized. The last two tasks, reach-hard and peg-insert-side-hard, are custom-made; there, the green goal and the green peg are randomly initialized either to the left or to the right of the gripper with equal probability, and they are not initially visible to the hand-centric perspective. Because of the severely limited hand-centric observability, the third-person perspective is crucial for learning to direct the gripper to the correct location. This is especially the case for the reach-hard task, which we modified to prohibit any vertical end-effector movement. See Section C.1 in the Appendix for more details about each task.

Hyperparameters that are common to all agents are shared for a given task. With agents that include regularization, we tune the regularization weight(s) on a validation sample from the test distribution. Test success rate is computed on a separate sample of 20 environments from the test distribution.

## 4.3 META-WORLD RESULTS AND DISCUSSION

Main experimental results in Meta-World are summarized in Table 3. Figure 6 provides detailed comparisons between the four DrQ-v2 agents introduced above. When using both perspectives, regularizing the third-person perspective's representation via a VIB reduces the interquartile mean of the out-of-distribution failure rate across all six tasks by 64% (relative). We also note that this method achieves the best performance in each individual task, albeit sometimes with less sample efficiency. To properly explain these phenomena, we now embark on a more stratified analysis and discussion of the results.

Table 3: Aggregate out-of-distribution generalization performance across all six Meta-World tasks computed from taking the three highest success rates achieved in each run of each DrQ-v2 agent. Using both perspectives with a VIB on the third-person perspective's representation results in the best aggregate success rate. The last three agents are ablations of the above and are presented and discussed in Appendix C.4.

| | success rate (%) | | | |
| --- | --- | --- | --- | --- |
| | mean | median | IQM | 95% CI of IQM |
| $\pi \circ O_{h+p}$ | 69.6 | 73.1 | 73.7 | $[72.1, 75.4]$ |
| $\pi \circ O_{3+p}$ | 55.6 | 51.7 | 56.6 | $[53.4, 59.1]$ |
| $\pi \circ O_{h+3+p}$ | 67.4 | 68.1 | 66.3 | $[62.9, 69.8]$ |
| $\pi \circ O_{h+3+p} + \text{VIB}(\mathbf{z}_3)$ | **84.4** | **82.5** | **87.7** | $[\mathbf{85.2, 90.0}]$ |
| $\pi \circ O_{h+3+p} + \text{VIB}(\mathbf{z}_h) + \text{VIB}(\mathbf{z}_3)$ | 51.8 | 54.7 | 58.2 | $[50.5, 65.9]$ |
| $\pi \circ O_{3'+3+p} + \text{VIB}(\mathbf{z}_3)$ | 40.6 | 32.8 | 34.5 | $[32.7, 36.4]$ |
| $\pi \circ O_{h+3+p} + \ell_2(\mathbf{z}_3)$ | 70.3 | 70.0 | 69.8 | $[67.7, 72.3]$ |

**Characterization of hand-centric observability via training performance.** When the world state is sufficiently observable via the hand-centric perspective, we expect the convergence during training of $\pi \circ O_{h+p}$ to match or surpass that of $\pi \circ O_{3+p}$. We find that this is indeed the case for handle-press-side, button-press, soccer, and peg-insert-side (high and moderate hand-centric observability), and not the case for reach-hard or peg-insert-side-hard (low hand-centric observability). This validates our selection and framing of the tasks at different levels of hand-centric observability. Interestingly, we observe that in peg-insert-side-hard, $\pi \circ O_{h+p}$ eventually achieves some success during training by "zooming out" to improve its observability.

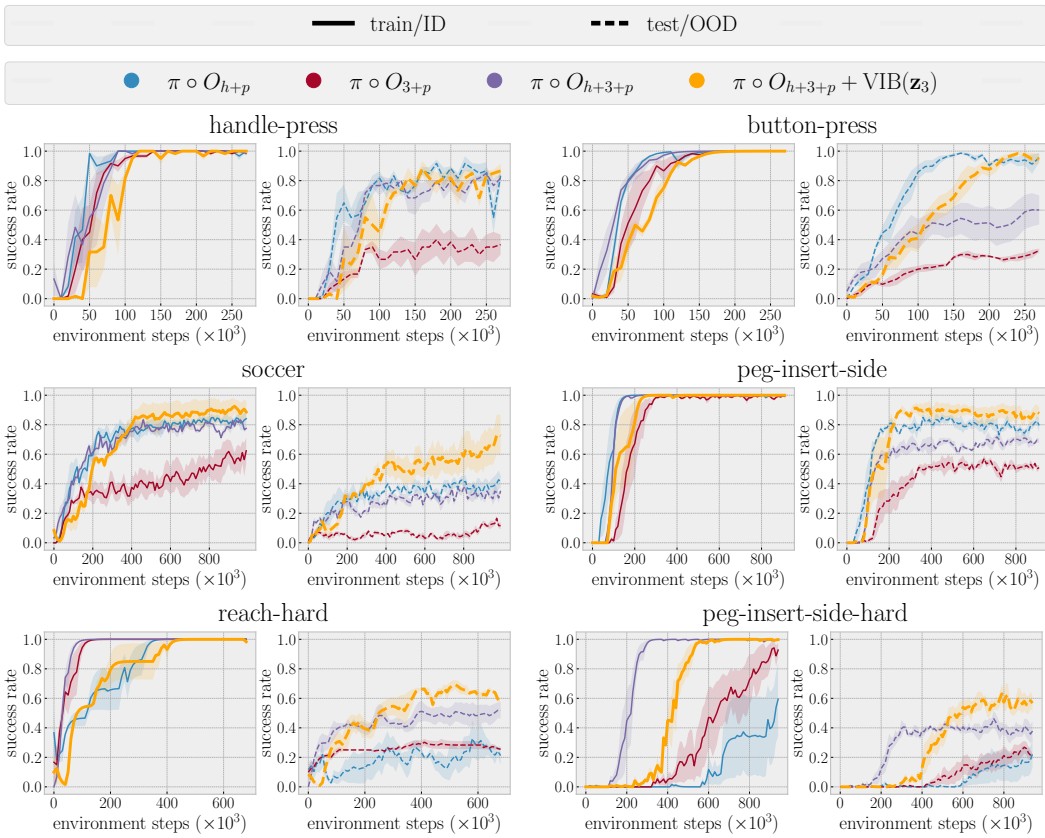

Figure 6: DrQ-v2 results for Meta-World. Each row contains results for two manipulation tasks that roughly exhibit the same level of hand-centric observability, which decreases from top to bottom (high, moderate, low). Using the proposed approach (both perspectives with a VIB on the third-person perspective's representation) leads to the best out-of-distribution generalization performance for all levels of hand-centric observability (though it is matched by the hand-centric perspective when hand-centric observability is high, as expected). Shaded regions indicate the standard error of the mean over three random seeds.

**Hand-centric perspective vs. third-person perspective.** When hand-centric observability is high or moderate, $\pi \circ O_{h+p}$ generalizes better out-of-distribution than $\pi \circ O_{3+p}$, corroborating results from Section 3 with another form of distribution shift. When hand-centric observability is low, $\pi \circ O_{h+p}$ both trains and generalizes worse than $\pi \circ O_{3+p}$. This supports our motivation for considering using both perspectives in conjunction.

**Effect of combining the hand-centric and third-person perspectives.** When hand-centric observability is high or moderate, including the third-person perspective can harm generalization. We see that for button-press, peg-insert-side, handle-press, and soccer, $\pi \circ O_{h+3+p}$ is sandwiched between $\pi \circ O_{h+p}$ and $\pi \circ O_{3+p}$ on the test distribution. The drop from $\pi \circ O_{h+p}$ to $\pi \circ O_{h+3+p}$ is significant for the former two tasks, and marginal for the latter two. This validates our hypothesis that including $O_3$ enables the agent to "overfit" to training conditions. When hand-centric observability is low, combining both perspectives results in $\pi \circ O_{h+3+p}$ matching or surpassing the training performance of $\pi \circ O_{h+p}$ and $\pi \circ O_{3+p}$, and greatly outperforming both at test time. This validates our hypothesis that, when necessary, including third-person observations helps resolve training difficulties arising from insufficient hand-centric observability.

**Effect of regularizing the third-person information stream via a VIB.** $\pi \circ O_{h+3+p} + \text{VIB}(\mathbf{z}_3)$ consistently improves upon $\pi \circ O_{h+3+p}$ in out-of-distribution generalization for all tasks except handle-press-side, in which the two are about equal. This directly indicates the benefit of the VIB regularization. These gains come at the cost of slightly delaying the convergence of training. However, it is arguable that this is inevitable and even desirable. A known phenomenon in neural network training is that spurious correlations or "shortcuts" in the data are sometimes easier to learn than causal relationships (Sagawa et al., 2019). Slower training and higher generalization may indicate

the avoidance of such behavior. Additionally, in button-press, $\pi \circ O_{h+3+p} + \text{VIB}(\mathbf{z}_3)$ recovers the out-of-distribution generalization exhibited by $\pi \circ O_{h+p}$, and when hand-centric observability is moderate, $\pi \circ O_{h+3+p} + \text{VIB}(\mathbf{z}_3)$ improves upon $\pi \circ O_{h+p}$.

**Ablations on $\pi \circ O_{h+3+p} + \text{VIB}(\mathbf{z}_3)$.** We conduct three ablations on this best-performing agent to better understand the design decisions underlying its gains. See Appendix C.4 for description, results, and discussion.

## 5 RELATED WORK

**Learning for vision-based object manipulation.** A wide range of works have focused on algorithmic development for end-to-end learning of vision-based object manipulation skills (Levine et al., 2016; Agrawal et al., 2016; Finn et al., 2016; 2017; Kalashnikov et al., 2018; Srinivas et al., 2018; Ebert et al., 2018; Zhu et al., 2018; Jayaraman et al., 2018; Rafailov et al., 2021). Some works on learned visuomotor control use eye-in-hand cameras for tasks such as grasping (Song et al., 2020) and insertion (Zhao et al., 2020; Puang et al., 2020; Luo et al., 2021; Valassakis et al., 2021), and others which pre-date end-to-end visuomotor learning use both eye-in-hand and third-person cameras for visual servoing (Flandin et al., 2000; Lippiello et al., 2005). Very few works consider the design of camera placements (Zaky et al., 2020) or conduct any controlled comparisons on different combinations of visual perspectives (Zhan et al., 2020; Mandlekar et al., 2021; Wu et al., 2021). Unlike all of these works, we propose specific hypotheses regarding the benefits of different choices of visual perspective and perform a systematic empirical validation of these hypotheses with evaluation on multiple families of learning algorithms, manipulation tasks, and distribution shifts. Concurrently with our work, Jangir et al. (2022) investigate fusing information from hand-centric and third-person perspectives using a cross-view attention mechanism and demonstrate impressive sim2real transfer.

**The role of perspective on generalization.** Hill et al. (2019) assess an agent learning to execute language instructions in simulated environments using high-level actions and find that using an egocentric observation space results in better systematic generalization to new instruction noun-verb combinations. Szot et al. (2021) find that an agent tasked to pick up a certain object (using abstracted grasping) in a cluttered room generalizes better to unseen objects and room layouts when using wrist- and head-mounted cameras in conjunction. Our work provides complementary evidence for the effect of perspective on the generalization of learned agents in a markedly different setting: we consider vision-based physical manipulation. Also, the aforementioned works rely on memory-augmented agents to resolve partial observability as is common in navigation tasks, whereas we use third-person observations as is standard in tabletop manipulation and demonstrate the importance of regularizing their representation.

**Invariances through data augmentation in reinforcement learning.** Several works have investigated ways to apply standard data augmentation techniques from computer vision in the reinforcement learning setting (Laskin et al., 2020; Kostrikov et al., 2020; Yarats et al., 2021). These works consider data augmentation as a means to prescribe invariances to image-space transformations, whereas we are concerned with how different observation functions facilitate generalization to environmental transformations. To emphasize that these directions are orthogonal, we use DrQ (Kostrikov et al., 2020) and DrQ-v2 (Yarats et al., 2021) in our experiments.

## 6 CONCLUSION

In this work, we abstain from algorithm development and focus on studying an underexplored design choice in the embodied learning pipeline: the observation function. While hand-centric robotic perception is more traditionally instrumented with tactile sensing, our findings using vision affirm that perspective, even when controlling for modality, can play an important role in learning and generalization. This insight may very well apply to robotic systems that leverage tactile sensing. Overall, in the context of end-to-end learning for visuomotor manipulation policies, our findings lead us to recommend using hand-centric perspectives when their limited observability is sufficient, and otherwise defaulting to using both hand-centric and third-person perspectives while regularizing the representation of the latter. The breadth of the learning algorithms, manipulation tasks, and distribution shifts that we base these conclusions on, coupled with their simplicity and lack of restrictive assumptions, suggests that these recommendations should be broadly applicable, even to more complex, longer-horizon tasks that feature sub-tasks analogous to those we experiment with.

ACKNOWLEDGMENTS

We thank Kaylee Burns, Ashvin Nair, Eric Mitchell, Rohan Taori, Suraj Nair, Ruohan Zhang, Michael Lingelbach, Qian Huang, Ahmed Ahmed, and Tengyu Ma for insightful discussions and feedback on early drafts. We also thank our anonymous ICLR reviewers for their constructive comments. This work was in part supported by Google, Apple, Stanford Institute for Human-Centered AI (HAI), Amazon Research Award (ARA), Autodesk, Bosch, Salesforce, and ONR grant N00014-21-1-2685. KH was supported by a Sequoia Capital Stanford Graduate Fellowship. CF is a fellow in the CIFAR Learning in Machines and Brains program.

REPRODUCIBILITY STATEMENT

Appendices A, B, and C flesh out the full experimental protocol in stringent detail. We expect this to be sufficient for independent replication of our main findings. Separately, we have included links to code used for our simulation experiments on our project website.

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

# A  Cube Grasping Experiment Details

## A.1  Environment Details

For the cube grasping experiments in Section 3, we investigate three types of distribution shifts. The experiment variants for DAgger and DrQ are summarized in Table 4. The DAC experiments featured a subset of these conditions explained in the caption of Figure 3. Figures 7, 8, and 9 visualize each type of distribution shift.

Table 4: Variants of the cube grasping environment with distribution shifts used for the DAgger and DrQ experiments. All task variants include initial object position and initial end-effector position randomization that is consistent across train and test. Table textures are from the describable textures dataset (DTD) (Cimpoi et al., 2014).

|  | train | test |
|---|---|---|
| table height | $z_{\text{shift}} = 0$ | $z_{\text{shift}} \in \{-0.10, -0.05, +0.05, +0.10\}$ |
| distractor objects | 1 red, 1 green, 1 blue | 3 of color $\in \{$red, green, blue, brown, white, black$\}$ |
| table texture | texture $\in$ 5 DTD textures | texture $\in$ 20 held-out DTD textures |

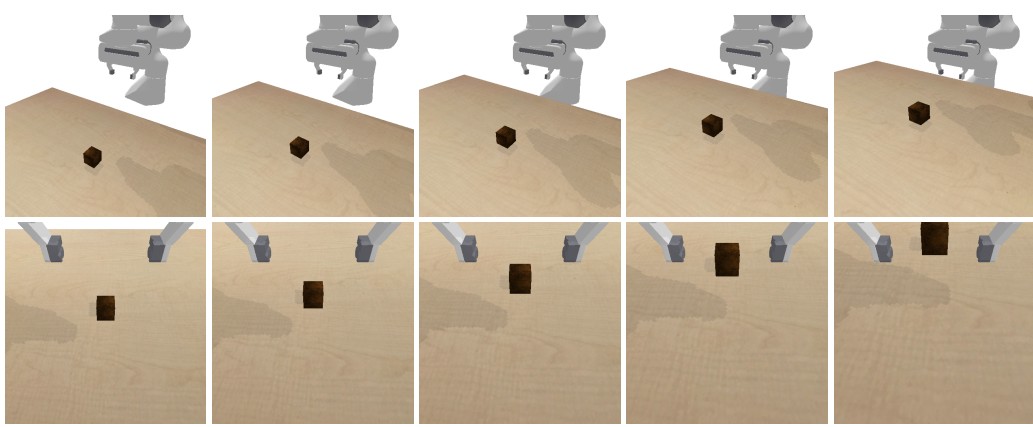

Figure 7: Visualization of the table height distribution shift used in the cube grasping experiments. From left to right, $z_{\text{shift}}$ is $-0.10, -0.05, 0, +0.05, +0.10$. The top and bottom rows contain the third-person and hand-centric perspectives, respectively. Positions of the cube and end-effector are not randomized in this visualization for the sake of clarity.

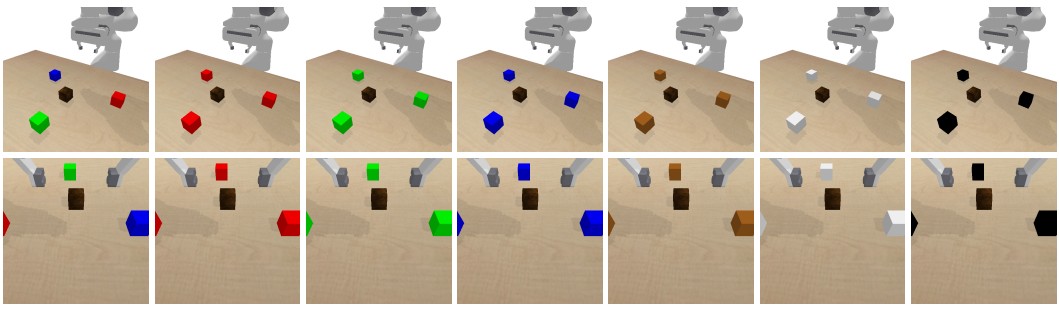

Figure 8: Visualization of the distractor objects distribution shift used in the cube grasping experiments. From left to right, we have "mix" (1 red, 1 green, 1 blue), 3 red, 3 green, 3 blue, 3 brown, 3 white, and 3 black. The top and bottom rows contain the third-person and hand-centric perspectives, respectively. Positions of the cubes and end-effector are not randomized in this visualization for the sake of clarity.

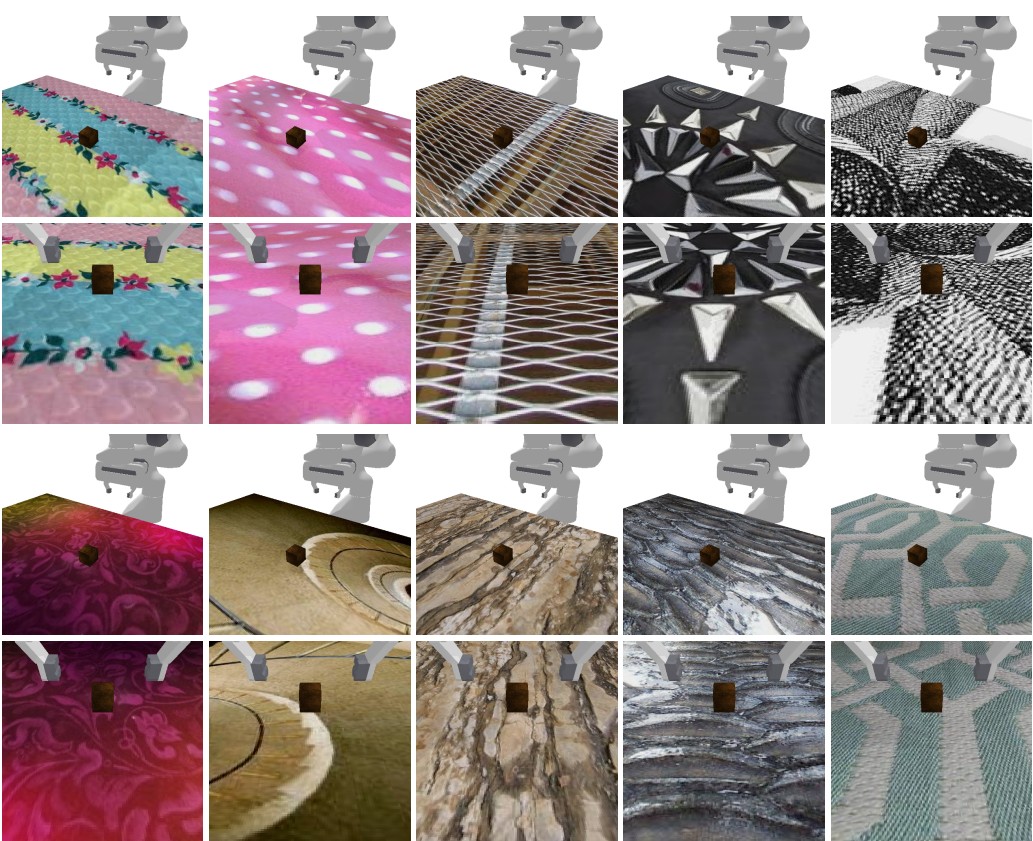

Figure 9: Visualization of the table textures distribution shift used in the cube grasping experiments. The top two rows contain the third-person and hand-centric perspectives of the five table textures used during training for DAgger and DrQ, and the bottom two rows contain the perspectives of five held-out textures used at test time (out of twenty total held-out textures). Positions of the cube and end-effector are not randomized in this visualization for the sake of clarity. The textures were acquired from the describable textures dataset (DTD) (Cimpoi et al., 2014).

## A.2 ALGORITHMS

The dataset aggregation (DAgger) algorithm proposed by Ross et al. (2011) is an iterative online algorithm for training an imitation learning policy. In each iteration $i$ (which we call a "DAgger round"), the current policy $\pi_i$ is run to sample a set of trajectories, and an expert policy $\pi^*$ is used to label each of the visited states with an optimal action. These labeled trajectories are aggregated into a dataset $\mathcal{D}$ that grows in size over the DAgger rounds, and the imitation learning policy $\hat{\pi}_i$ is trained on the entire $\mathcal{D}$ for some number of epochs before repeating the above procedure in the next iteration. The trajectory-generating policy $\pi_i$ is often modified such that in earlier DAgger rounds the expert policy $\pi^*$ is utilized more heavily than the imitation learning policy $\hat{\pi}_i$ when collecting new trajectories, i.e. $\pi_i = \beta_i \pi^* + (1 - \beta_i)\hat{\pi}_i$, where $\beta_i$ is typically annealed over time (e.g., linearly from 1 to 0 over the DAgger rounds).

The Data-regularized Q (DrQ) algorithm proposed by Kostrikov et al. (2020) is a model-free, off-policy, actor-critic reinforcement learning algorithm that applies image augmentation techniques commonly used in computer vision (primarily random shifts) to input images, along with regularizations of the $Q$ target and function, such that deep neural network-based agents can be trained effectively from pixels. The original DrQ paper uses soft actor-critic (Haarnoja et al., 2018) and DQN (Mnih et al., 2013) as backbones; we use the soft actor-critic version in our experiments because the cube grasping action space is continuous.

The discriminator actor-critic algorithm (DAC) was proposed in Kostrikov et al. (2018) and is an off-policy version of the generative adversarial imitation learning (GAIL) method (Ho & Ermon, 2016). Unlike Kostrikov et al. (2018) we use a deterministic reinforcement learning algorithm similar to that of Fujimoto et al. (2018), as we find this helps stability. To scale the method to image observations, we apply similar augmentation techniques as in Kostrikov et al. (2020).

## A.3 MODEL ARCHITECTURES

For DAgger in the cube grasping experiments discussed in Section 3, we feed the $84 \times 84$ images into a ResNet-18 convolutional image encoder (He et al., 2016) trained from scratch, with the final classification layer replaced by a linear layer that outputs a 64-dimensional representation. We concatenate proprioceptive information (3D end-effector position relative to the robot base, 1D gripper width, and a Boolean contact flag for each of two gripper "fingers") to the image representation, and the result is passed into feedforward policy and value networks with two hidden layers of 32 units each.

For DrQ, we use the original actor-critic DrQ model proposed by Kostrikov et al. (2020), except for one modification: we concatenate proprioceptive information (3D end-effector position relative to the robot base, 1D gripper width, and a Boolean contact flag for each of two gripper "fingers") to the flattened image representation before feeding it into the actor and critic networks.

For the DAC algorithm we use the same convolutional architectures as Kostrikov et al. (2018). The convolutional encoder is shared between the discriminator, actor and critic. We use additional MLP heads with capacities 128, 256 and 256 respectively for those components, as we empirically found that lower-capacity networks decrease the likelihood of overfitting to spurious features.

## A.4 HYPERPARAMETERS

The DAgger, DrQ, and DAC hyperparameters used in the cube grasping experiments are listed in Tables 5, 6, and 7, respectively.

Table 5: The default hyperparameters used for DAgger in the cube grasping environment.

| parameter | setting |
|---|---|
| num. DAgger rounds | 6 |
| num. new episodes per round | 200 |
| num. epochs per round | 15 |
| batch size | 256 |
| $\beta$ schedule (% time expert policy is used) | linear anneal from 1 to 0 over 6 rounds |
| learning rate | $10^{-3}$ |

Table 6: The default hyperparameters used for DrQ in the cube grasping environment.

| parameter | setting |
|---|---|
| replay buffer capacity | $10^5$ |
| action repeat | 2 |
| seed steps | 1000 |
| $n$-step returns | 3 |
| mini-batch size | 128 |
| discount $\gamma$ | 0.99 |
| optimizer | Adam |
| learning rate | $10^{-3}$ |
| critic target update frequency | 2 |
| critic Q-function soft-update rate $\tau$ | 0.01 |
| actor update frequency | 2 |
| actor log stddev. bounds | $[-10, 2]$ |
| init. temperature | 0.1 |
| features dim. | 50 |
| hidden dim. | 1024 |

Table 7: The default hyperparameters used for DAC in the cube grasping environment.

| parameter | setting |
|---|---|
| replay buffer capacity | $15 \times 10^3$ |
| action repeat | 1 |
| seed steps | 200 |
| mini-batch size | 512 |
| discount $\gamma$ | 0.99 |
| optimizer | Adam |
| learning rate | $10^{-3}$ |
| actor | deterministic |
| critic Q-function soft-update rate $\tau$ | 0.01 |
| features dim. | 64 |
| discriminator hidden dim. | 128 |
| actor hidden dim. | 256 |
| critic hidden dim. | 256 |

## A.5 Ablation Study: Removing the Data Augmentation in DrQ

In this experiment, we investigate the effect of the data augmentation component of the DrQ algorithm by ablating it. The motivation is to see whether data augmentation is still necessary for a policy using the hand-centric perspective, which already leads to lower overfitting and better generalization. The results in Figure 10 reveal that the augmentation is indeed still crucial because without it, training does not converge even with much more environment interaction. However, the hand-centric perspective does still enable the agent to make greater progress.

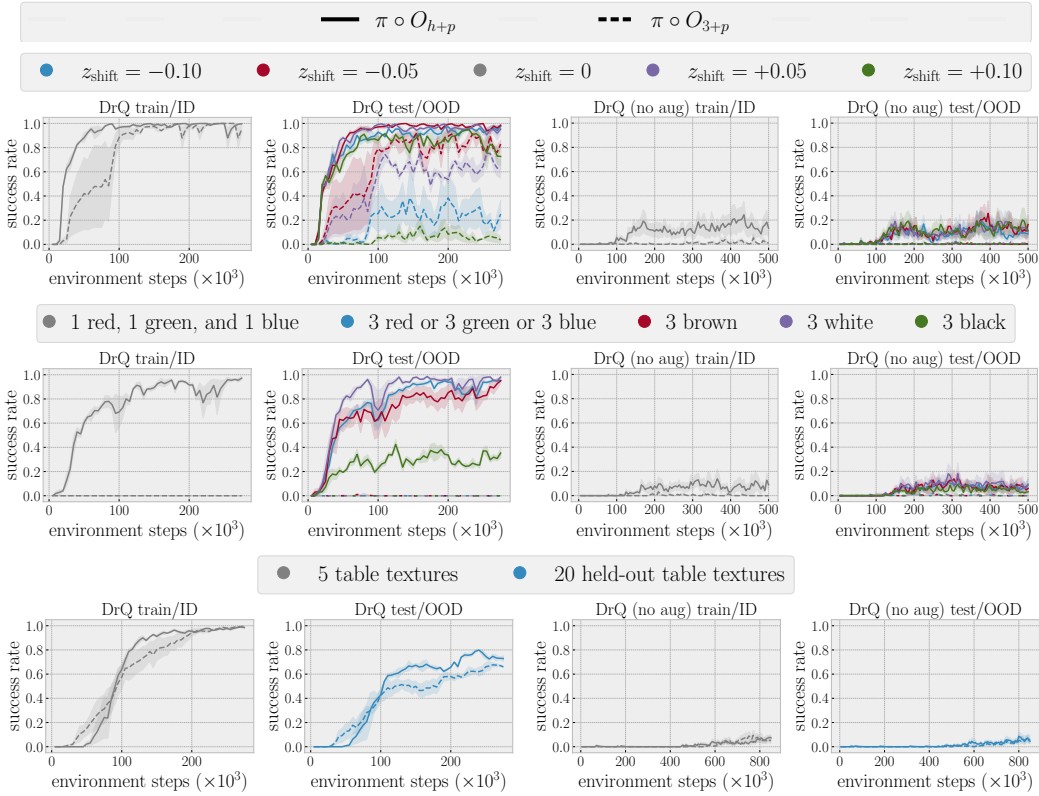

Figure 10: DrQ results for cube grasping with (left) and without (right) image augmentation. Note that the left half of this figure is an exact replica of the right half of Figure 2. Ablating the image augmentation component of DrQ reveals its importance; without it, training fails to converge even with much larger amounts of environment interaction. However, the hand-centric perspective still facilitates faster training than the third-person perspective for the first two experiment variants. Shaded regions indicate the standard error of the mean over three random seeds.

## A.6 Minor Discrepancies between Algorithms

Due to implementation idiosyncrasies, there are minor discrepancies in how each algorithm processes environment observations. Following Kostrikov et al. (2020), for DrQ and DAC-DrQ observations are "frame-stacked" with three time steps' observations. This was not done for DAgger. Proprioceptives are used for DAgger and DrQ but not for DAC-DrQ. We take the position that these differences increase the generalizability of the trends we observe. We emphasize that the target effect under consideration is $O_h$ vs. $O_3$ in each setting.

# B  REAL ROBOT EXPERIMENTS

In this section, we discuss real robot experiments resembling the simulated experiments in Section 3, which presented a head-to-head comparison between the hand-centric and third-person perspectives. A few minor differences exist between the simulated and real experiments, which are delineated in Section B.1. However, the key findings discussed in Section B.2 match those from the simulated experiments, validating the improved generalization performance that the hand-centric perspective provides over the third-person perspective in vision-based manipulation tasks.

## B.1  EXPERIMENTAL SETUP

As in the simulated experiments in Section 3, we conduct the real robot experiments with a Franka Emika Panda robot arm. The robot is tasked with grasping and lifting a sponge from a gray bin while other distractor objects are present. The action space is 4-DoF, consisting of 3-DoF end-effector position control and 1-DoF gripper control. Observation functions include $O_h$ and $O_3$, which output $100 \times 100$ RGB images, and $O_p$, which outputs 3D end-effector position relative to the robot base and 1D gripper width. As before, we perform a head-to-head comparison between $\pi \circ O_{h+p}$ and $\pi \circ O_{3+p}$, i.e. the policies using hand-centric and third-person visual perspectives (and proprioceptive observations), respectively.

During the training phase, we train a behavioral cloning policy until convergence using the same set of 360 demonstrations for both $\pi \circ O_{h+p}$ and $\pi \circ O_{3+p}$, collected via robot teleoperation using a virtual reality headset and controller. This is roughly the quantity of demonstrations needed to achieve reliable grasping performance on the training distribution (85% success rate over 20 episodes) due to randomized initial object positions as well as randomized initial gripper position. Unlike in Section 3, we do not use dataset aggregation (DAgger) here. The target object to grasp is a Scotch-Brite sponge, with the green side always facing upwards. In addition, at training time, three distractor objects are present: a folded red washcloth, a folded blue washcloth, and a yellow sponge decorated with spots.

At test time, we introduce three categories of distribution shifts, similar to those in Section 3: unseen table heights, unseen distractor objects, and unseen table textures. Figures 11, 12, and 13 illustrate these distribution shifts. When testing against unseen table heights and table textures, the scene contains the same set of target object and distractor objects that we used at training time.

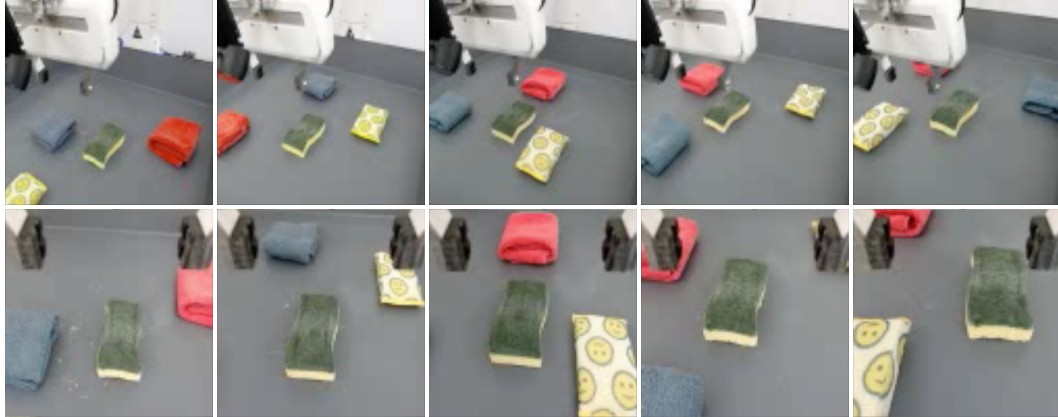

Figure 11: Table height distribution shifts. Columns from left to right: $-0.05$ m, $-0.025$ m, train, $+0.025$ m, $+0.05$ m. Top (bottom): observations from $O_3$ ($O_h$).

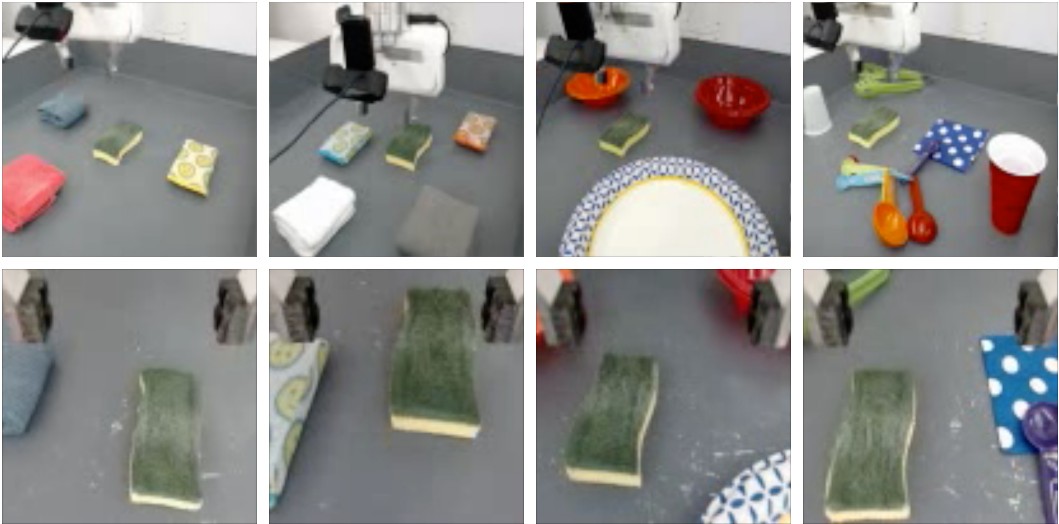

Figure 12: Distractor object distribution shifts. Columns from left to right: train, distractor test set 1, distractor test set 2, distractor test set 3. Top (bottom): observations from $O_3$ ($O_h$).

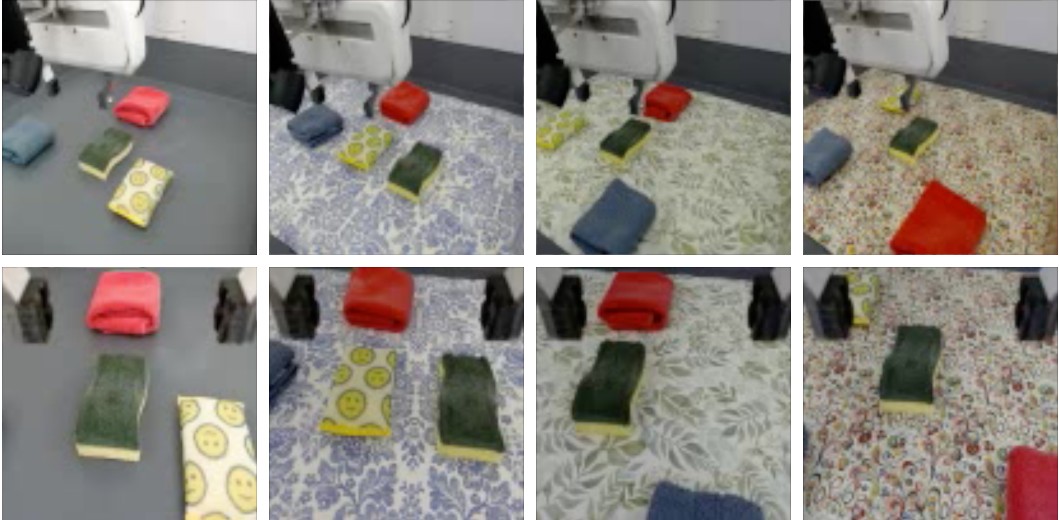

Figure 13: Table texture distribution shifts. Columns from left to right: train, blue floral, green watercolor garden, rainbow floral. Top (bottom): observations from $O_3$ ($O_h$).

### B.2 EXPERIMENTAL RESULTS AND DISCUSSION

The real robot behavioral cloning results are reported in Table 8. We find that the hand-centric perspective leads to significantly greater out-of-distribution generalization performance across all three experiment variants despite both hand-centric and third-person policies achieving the same performance on the training distribution (85% success rate over 20 episodes), validating the results we see in simulation.

Table 8: Out-of-distribution generalization performance comparisons between a behavior cloning (BC) policy using a hand-centric visual perspective and a BC policy using a third-person perspective. The distribution shifts include unseen table heights (where $z_{shift}$ indicates the change from the base table height used during training, in meters), unseen distractor objects, and unseen table textures. Compared to the third-person perspective, the hand-centric perspective leads to better out-of-generalization distribution performance across all three distribution shifts. In-distribution results are provided to give an idea of a performance ceiling. Each success rate is computed over 20 evaluation episodes.

|  |  | success rate (%) | |
| --- | --- | --- | --- |
|  |  | $\pi \circ O_{h+p}$ | $\pi \circ O_{3+p}$ |
| unseen table heights | $z_{shift} = -0.05$ m | **50** | 10 |
|  | $z_{shift} = -0.025$ m | **80** | 60 |
|  | $z_{shift} = +0.025$ m | **85** | 35 |
|  | $z_{shift} = +0.05$ m | **65** | 0 |
| unseen distractor objects | distractor test set 1 | **55** | 40 |
|  | distractor test set 2 | **55** | 10 |
|  | distractor test set 3 | **45** | 20 |
| unseen table textures | blue floral | **50** | 5 |
|  | green watercolor garden | **25** | 15 |
|  | rainbow floral | **10** | 5 |
| in-distribution |  | **85** | 85 |

# C    META-WORLD EXPERIMENT DETAILS

## C.1    INDIVIDUAL TASK DESCRIPTIONS

In this section, we explain the tasks that the agents must learn to accomplish in the six Meta-World environments discussed in Section 4.2 and visualized in Figure 5. We also explain why each task falls under a certain level of hand-centric observability. For details regarding the train and test distributions, see Appendix C.2.

- handle-press-side: The goal is to press the handle fully downwards. Hand-centric observability is high because the handle is well aligned with the hand-centric camera's field of view.

- button-press: The goal is to push the button fully inwards. Hand-centric observability is high because the button is well in view of the hand-centric camera, and the button remains largely in view as the gripper approaches and presses it.

- soccer: The goal is to push or pick-and-place the ball into the center of the goal net. Hand-centric observability is moderate because when the gripper approaches the ball, the observability of the goal net is appreciably reduced.

- peg-insert-side: The goal is to lift the peg and insert it into the hole in the target box. Hand-centric observability is moderate because when the gripper approaches the peg, the observability of the target box is appreciably reduced.

- reach-hard: The goal is to move the gripper to the green goal site, which is initialized either to the left or right side of the gripper with equal probability (see Figure 14). Hand-centric observability is low because the gripper is initialized at the same height as the goal, and we restrain the gripper from moving vertically. Effectively, if given just the hand-centric perspective's observations, the agent does not know in which direction to move the gripper in the beginning of an episode.

- peg-insert-side-hard: The goal is the same as in peg-insert-side, but like the green goal site in reach-hard, the peg in this environment is initialized either to the left or right side of the gripper with equal probability (see Figure 14). Hand-centric observability is low because the gripper is initialized at the same height as the peg such that the peg is not initially visible to the hand-centric view (though we do *not* prohibit vertical movement of the gripper as in reach-hard, since this would make the peg insertion part of the task impossible), and also because the peg and target box are initialized much farther apart than they are in peg-insert-side (thus, the target box is completely out of view as the agent approaches and grasps the peg).

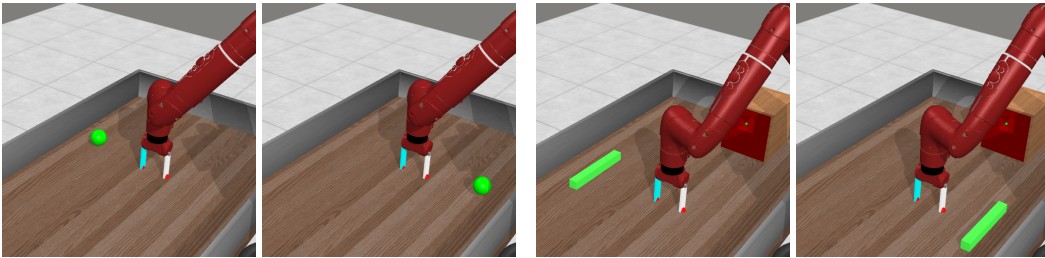

Figure 14: Visualizations of the two sides that the green goal site or peg can be initialized to in the reach-hard and peg-insert-side-hard tasks in Meta-World, respectively. One of the two sides is chosen via "coin flip" at the beginning of each episode.

## C.2 TRAIN AND TEST DISTRIBUTIONS

At training time, initial positions of the objects in the Meta-World tasks are uniformly sampled within some support. At test time, initial positions are sampled from a uniform distribution that is completely disjoint from the training distribution, such that we test on out-of-distribution initial object positions. To implement this, at test time we resample the set of initial object positions if *any* of the positions overlaps with its train-time distribution. The full set of train-time and test-time initial object positions is shown in Table 9. For visualizations, see Figure 15.

Table 9: The distributions of the initial object positions used in the six Meta-World environments. At training time, we use the "narrow" initial object positions. At test time, we use the set difference between the "narrow" and "wide" initial object positions, i.e. we repeatedly resample from "wide" until none of the objects in the environment lie in the "narrow" distribution. As a result, the train- and test-time distributions are disjoint. The range of positions for each object is given as a pair of coordinates: $(x_{\text{low}}, y_{\text{low}}, z_{\text{low}})$ and $(x_{\text{high}}, y_{\text{high}}, z_{\text{high}})$.

| task | "narrow" init. obj. positions $(x, y, z)$ | "wide" init. obj. positions $(x, y, z)$ |
|---|---|---|
| handle-press-side | $\text{handle}_{\text{low}} : (-0.35, 0.55, -0.001)$
$\text{handle}_{\text{high}} : (-0.15, 0.65, 0.001)$ | $\text{handle}_{\text{low}} : (-0.55, 0.4, -0.001)$
$\text{handle}_{\text{high}} : (-0.15, 0.8, 0.001)$ |
| button-press | $\text{button}_{\text{low}} : (-0.2, 0.85, 0.115)$
$\text{button}_{\text{high}} : (0, 0.9, 0.115)$ | $\text{button}_{\text{low}} : (-0.4, 0.75, 0.115)$
$\text{button}_{\text{high}} : (0.2, 0.9, 0.115)$ |
| soccer | $\text{ball}_{\text{low}} : (-0.2, 0.6, 0.03)$
$\text{ball}_{\text{high}} : (0., 0.7, 0.03)$
$\text{goal}_{\text{low}} : (-0.2, 0.8, 0.0)$
$\text{goal}_{\text{high}} : (0., 0.9, 0.0)$ | $\text{ball}_{\text{low}} : (-0.3, 0.6, 0.03)$
$\text{ball}_{\text{high}} : (0.1, 0.7, 0.03)$
$\text{goal}_{\text{low}} : (-0.3, 0.8, 0.0)$
$\text{goal}_{\text{high}} : (0.1, 0.9, 0.0)$ |
| peg-insert-side | $\text{peg}_{\text{low}} : (.05, 0.55, 0.02)$
$\text{peg}_{\text{high}} : (.15, 0.65, 0.02)$
$\text{goal}_{\text{low}} : (-0.325, 0.5, -0.001)$
$\text{goal}_{\text{high}} : (-0.275, 0.6, 0.001)$ | $\text{peg}_{\text{low}} : (.0, 0.5, 0.02)$
$\text{peg}_{\text{high}} : (.2, 0.7, 0.02)$
$\text{goal}_{\text{low}} : (-0.35, 0.4, -0.001)$
$\text{goal}_{\text{high}} : (-0.25, 0.7, 0.001)$ |
| reach-hard | if goal is on left of gripper:
    $\text{goal}_{\text{low}} : (-0.2, 0.85, 0.05)$
    $\text{goal}_{\text{high}} : (-0.2, 0.85, 0.05)$
if goal is on right of gripper:
    $\text{goal}_{\text{low}} : (-0.2, 0.35, 0.05)$
    $\text{goal}_{\text{high}} : (-0.2, 0.35, 0.05)$ | if goal is on left of gripper:
    $\text{goal}_{\text{low}} : (-0.5, 0.9, 0.05)$
    $\text{goal}_{\text{high}} : (0.1, 0.9, 0.05)$
if goal is on right of gripper:
    $\text{goal}_{\text{low}} : (-0.5, 0.3, 0.05)$
    $\text{goal}_{\text{high}} : (0.1, 0.3, 0.05)$ |
| peg-insert-side-hard | if peg starts on left of gripper:
    $\text{peg}_{\text{low}} : (-0.025, 0.85, 0.02)$
    $\text{peg}_{\text{high}} : (0.025, 0.85, 0.02)$
    $\text{goal}_{\text{low}} : (-0.525, 0.5, -0.001)$
    $\text{goal}_{\text{high}} : (-0.525, 0.6, 0.001)$
if peg starts on right of gripper:
    $\text{peg}_{\text{low}} : (-0.025, 0.35, 0.02)$
    $\text{peg}_{\text{high}} : (0.025, 0.35, 0.02)$
    $\text{goal}_{\text{low}} : (-0.525, 0.5, -0.001)$
    $\text{goal}_{\text{high}} : (-0.525, 0.6, 0.001)$ | if peg starts on left of gripper:
    $\text{peg}_{\text{low}} : (-0.3, 0.85, 0.02)$
    $\text{peg}_{\text{high}} : (0.1, 0.85, 0.02)$
    $\text{goal}_{\text{low}} : (-0.525, 0.3, -0.001)$
    $\text{goal}_{\text{high}} : (-0.525, 0.75, 0.001)$
if peg starts on right of gripper:
    $\text{peg}_{\text{low}} : (-0.3, 0.35, 0.02)$
    $\text{peg}_{\text{high}} : (0.1, 0.35, 0.02)$
    $\text{goal}_{\text{low}} : (-0.525, 0.3, -0.001)$
    $\text{goal}_{\text{high}} : (-0.525, 0.75, 0.001)$ |

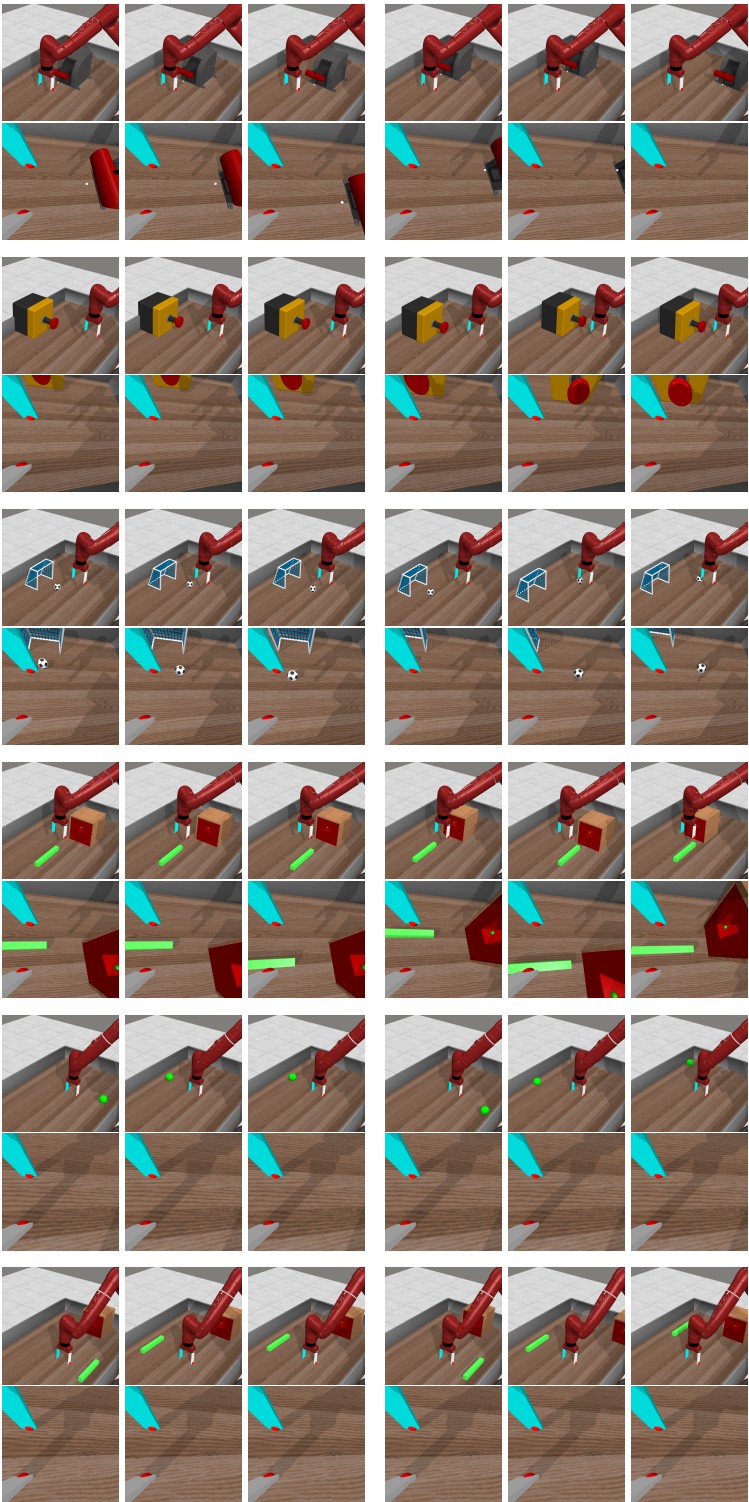

Figure 15: Visualization of the train and test distributions in the six Meta-World environments used in the experiments in Section 4. The three columns in the left half of the figure show three sets of initial object positions randomly sampled from the training distribution; the three columns in the right half correspond to the test distribution. From top to bottom are handle-press-side, button-press, soccer, peg-insert-side, reach-hard, and peg-insert-side-hard. Both the third-person and hand-centric perspectives are shown for each random initialization.

## C.3   DRQ-V2

DrQ-v2 (Yarats et al., 2021) is a state-of-the-art vision-based actor-critic reinforcement learning algorithm that uses deep deterministic policy gradients (DDPG) (Lillicrap et al., 2015) as a backbone (whereas DrQ-v1 by Kostrikov et al. (2020) uses soft actor-critic). The DrQ-v2 model includes:

- a convolutional image encoder $f_\xi$ that outputs representation $\mathbf{z} = f_\xi(\mathrm{aug}(\mathbf{o}))$ given frame-stacked image observations $\mathbf{o}$ and a data augmentation function $\mathrm{aug}$,

- two critic networks $Q_{\theta_k}$ that output Q-values $Q_{\theta_k}(\mathbf{z}, \mathbf{a}), k = 1, 2$, à la clipped double Q-learning (Fujimoto et al., 2018),

- and an actor network $\pi_\phi$ that outputs action $\mathbf{a} = \pi_\phi(\mathbf{z}) + \epsilon, \epsilon \sim \mathcal{N}(0, \sigma^2)$, with $\sigma^2$ annealed over the course of training.

The individual critic losses are given by

$$\mathcal{L}_k = \mathbb{E}_{\tau \sim \mathcal{D}} \left[ (Q_{\theta_k}(\mathbf{z}, \mathbf{a}) - y)^2 \right], \ k = 1, 2 \tag{2}$$

where $\tau = (\mathbf{o}_t, \mathbf{a}_t, r_{t:t+n-1}, \mathbf{o}_{t+n})$ is a sample from replay buffer $\mathcal{D}$ and $y$ is the temporal difference target estimated via $n$-step returns:

$$y = \sum_{i=0}^{n-1} \gamma^i r_{t+i} + \gamma^n \min_{k \in \{1,2\}} Q_{\bar{\theta}_k}(\mathbf{z}_{t+n}, \mathbf{a}_{t+n}) \tag{3}$$

for slow-moving critic weights $\bar{\theta}_1, \bar{\theta}_2$. We omit presentation of the actor loss as we do not need to modify it; in DrQ-v2, only gradients from the critic loss are used to update the weights of the encoder(s).

In terms of the model architecture used in the experiments discussed in Section 4, we use the original DrQ-v2 architecture, except for two modifications: first, we concatenate proprioceptive information (3D end-effector position and 1D gripper width) to the flattened image representation before feeding it into the actor and critic networks. Second, when using two perspectives at the same time (e.g., hand-centric and third-person), we use two separate image encoders that do not share weights. The two representations are concatenated together (along with the proprioceptive information) and fed into the actor and critic networks. The dimensionality of each encoder's output representation is preserved, thereby doubling the dimensionality of the final combined image representation.

## C.4   ABLATIONS ON $\pi \circ O_{h+3+p} + \text{VIB}(\mathbf{z}_3)$

To better understand what makes $\pi \circ O_{h+3+p} + \text{VIB}(\mathbf{z}_3)$ work the best, we conduct the following ablations. Figure 17 presents the train and test curves of the ablation experiments.

**What if both perspectives are regularized?** The ablation agent $\pi \circ O_{h+3+p} + \text{VIB}(\mathbf{z}_h) + \text{VIB}(\mathbf{z}_3)$ adds a separate VIB to the hand-centric information stream in an analogous manner to how the third-person perspective's representation is regularized (detailed in Section 4.1). We use the same $\beta_3$ for both and tune $\beta_h$. Note that setting $\beta_h = 0$ for $\pi \circ O_{h+3+p} + \text{VIB}(\mathbf{z}_h) + \text{VIB}(\mathbf{z}_3)$ recovers $\pi \circ O_{h+3+p} + \text{VIB}(\mathbf{z}_3)$ modulo stochasticity in $\mathbf{z}_h$, so we limit the lowest value $\beta_h$ can take to 0.01. We find that in no task does $\pi \circ O_{h+3+p} + \text{VIB}(\mathbf{z}_h) + \text{VIB}(\mathbf{z}_3)$ outperform $\pi \circ O_{h+3+p} + \text{VIB}(\mathbf{z}_3)$, validating our choice of only regularizing the third-person perspective's representation.

**Assessing the importance of the hand-centric perspective.** $\pi \circ O_{3'+3+p} + \text{VIB}(\mathbf{z}_3)$ uses a second third-person perspective $O_{3'}$ instead of the hand-centric perspective $O_h$. Visualizations from this additional third-person perspective are shown in Figure 16. We re-tune $\beta_3$ for this agent. We find that $\pi \circ O_{3'+3+p} + \text{VIB}(\mathbf{z}_3)$ performs significantly worse than $\pi \circ O_{h+3+p} + \text{VIB}(\mathbf{z}_3)$, affirming the benefit of using the hand-centric perspective in the multi-perspective setting.

**$\mathbf{z}_h$-dependent regularization of $\mathbf{z}_3$.** $\text{VIB}(\mathbf{z}_3)$ reduces the information contained in $\mathbf{z}_3$ without directly considering $\mathbf{z}_h$. With the ablation agent $\pi \circ O_{h+3+p} + \ell_2(\mathbf{z}_3)$, we consider a simple form of $\mathbf{z}_h$-dependent regularization of $\mathbf{z}_3$ in which we push $\mathbf{z}_3$ towards $\mathbf{z}_h$ by adding a weighted regularization term $\alpha_3 \|\mathbf{z}_3 - \text{stopgrad}(\mathbf{z}_h)\|_2^2$ to the DrQ-v2 critic objective. This approach seems promising given that $\pi \circ O_{h+3+p}$ consistently outperforms $\pi \circ O_{3+p}$ across all six tasks, suggesting that even in the midst of substantial partial observability, $\mathbf{z}_h$ may represent information in a useful and generalizable way. We tune $\alpha_3$. We find that $\pi \circ O_{h+3+p} + \ell_2(\mathbf{z}_3)$ marginally improves over vanilla $\pi \circ O_{h+3+p}$ but still comes far short of $\pi \circ O_{h+3+p} + \text{VIB}(\mathbf{z}_3)$, suggesting that the two perspectives contain important complementary information that is better represented separately.

| handle-press-side | button-press | soccer | peg-insert-side | reach-hard | peg-insert-side-hard |
|---|---|---|---|---|---|

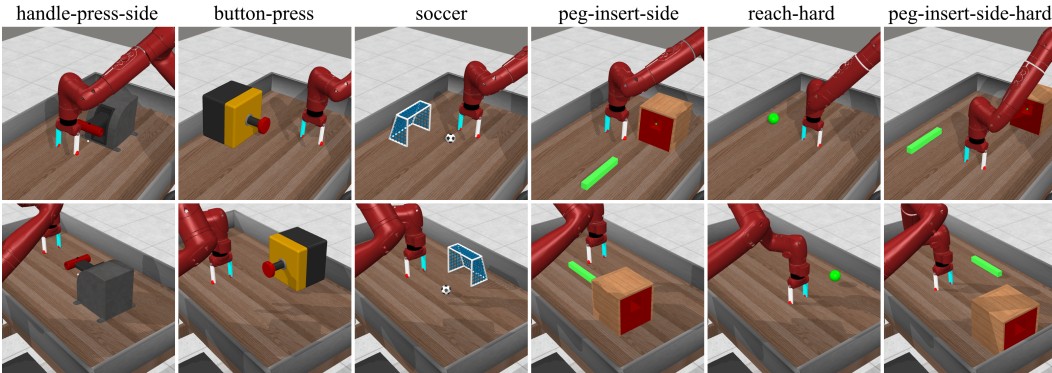

Figure 16: Visualizations of the two third-person perspectives used in the ablation agent $\pi \circ O_{3'+3+p} + \text{VIB}(\mathbf{z}_3)$ discussed in Section 4.3. The top row contains the original third-person perspective; the bottom row contains the second third-person perspective used only in the ablation.

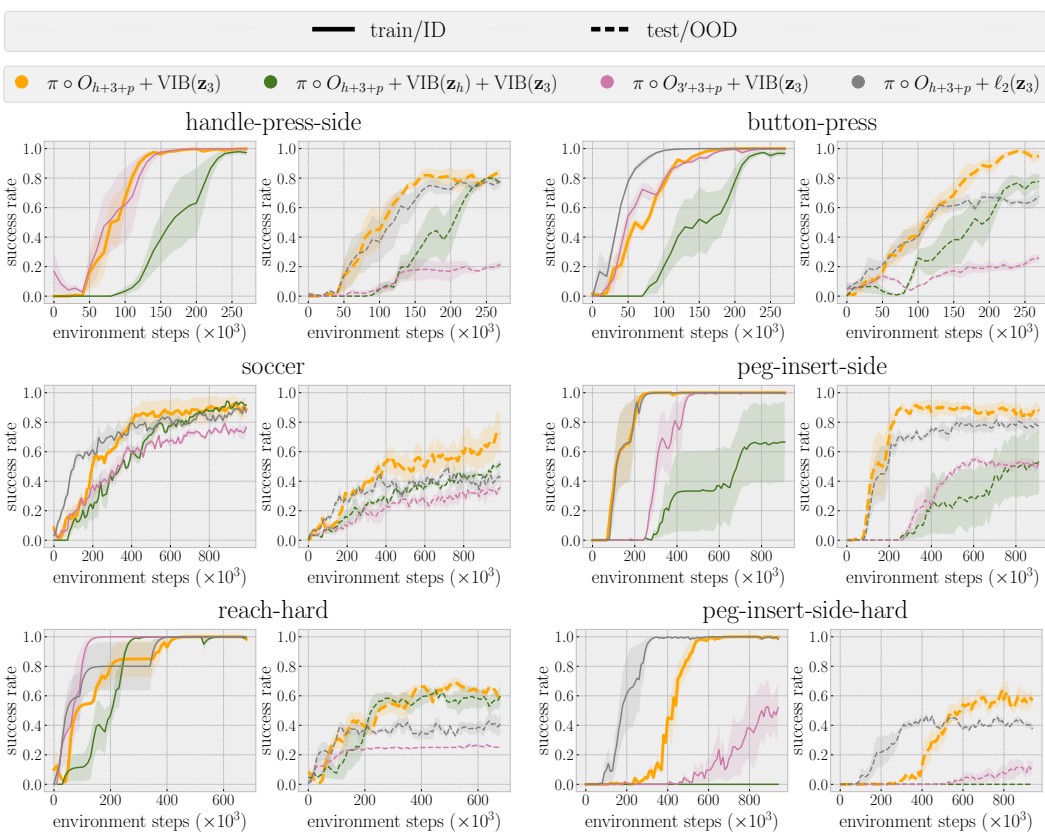

Figure 17: Ablation studies on $\pi \circ O_{h+3+p} + \mathrm{VIB}(\mathbf{z}_3)$ in Meta-World. Shaded regions indicate the standard error of the mean over three random seeds. Note that for $\pi \circ O_{h+3+p} + \mathrm{VIB}(\mathbf{z}_h) + \mathrm{VIB}(\mathbf{z}_3)$ in peg-insert-side, DrQ-v2 training did not converge within the specified number of training steps for one of three random seeds (hence the larger shaded regions and the lower out-of-distribution generalization performance). In addition, for $\pi \circ O_{h+3+p} + \mathrm{VIB}(\mathbf{z}_h) + \mathrm{VIB}(\mathbf{z}_3)$ in peg-insert-side-hard, DrQ-v2 training did not converge for any of the three seeds.

## C.5 PROPRIOCEPTION-ONLY ABLATION

In this ablation experiment, we demonstrate that visual observations are a necessary component of the observation space, i.e. that the tasks we experiment with cannot be consistently solved with proprioceptive observations alone. We run DrQ-v2 on all six Meta-World tasks introduced in Section 4.2 without image observations and show the results in Figure 18. Unlike policies that are afforded vision, these proprioception-only policies do not approach 100% success rate on the training distributions.

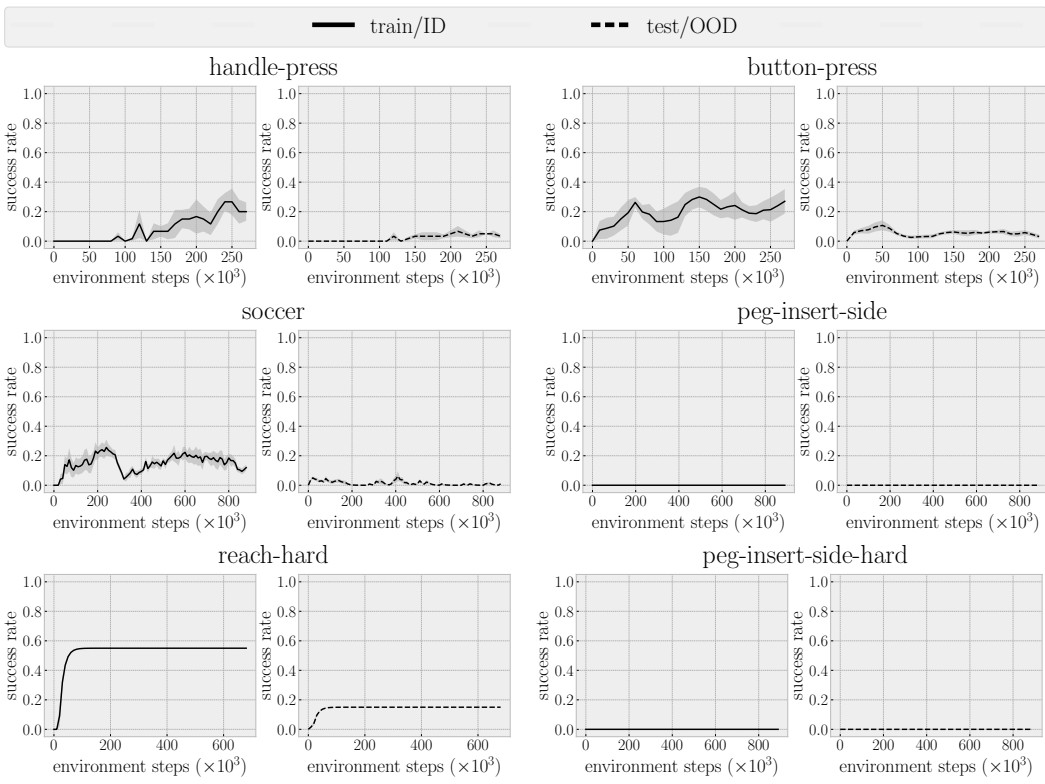

Figure 18: DrQ-v2 results for Meta-World for $\pi \circ O_p$, i.e. an agent that operates solely from proprioception. These results establish the necessity of vision for these tasks, as the policies are unable to consistently solve task instances without it. For the reach-hard task, in which the goal is randomly initialized to the left or right of the end-effector, the policy learns to always direct the end-effector towards one side. Shaded regions indicate the standard error of the mean over three random seeds.

C.6 EXPERIMENTS WITH END-EFFECTOR ORIENTATION CONTROL

The experiments in Section 4 involved a 4-DoF action space consisting of 3-DoF end-effector position control and 1-DoF gripper control, which was sufficient for solving all of the Meta-World tasks. In this section, we add one more degree of freedom for end-effector *orientation* control (allowing the parallel-jaw gripper to swivel) and then construct and experiment on two modified versions of the peg-insert-side task that cannot be solved without end-effector rotations. The train and test distributions of initial object center-of-masses are the same as those in the original peg-insert-side task.

In the first modified version, the end-effector is initially rotated 90 degrees from its original orientation, forcing the agent to rotate the end-effector before grasping the peg (see the center column of Figure 19 for a visualization). The second modified version of the task includes the following changes: (1) the proprioceptive observations also include the end-effector's orientation (as a quaternion), and (2) the peg—not the end-effector—is initially rotated by 90 degrees (see the rightmost column of Figure 19 for a visualization). Not only does (2) force the agent to rotate the end-effector before grasping the peg, but it also requires the agent to re-orient the peg correctly before inserting it into the box. The experimental results for DrQ-v2 in these two new environments are shown in Figure 20.

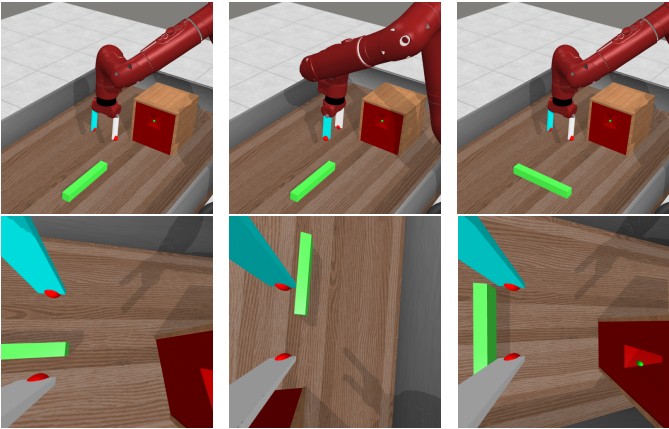

Figure 19: Visualizations of the two modified versions of the peg-insert-side task in Meta-World, where the first and second rows contain the third-person and hand-centric perspectives of the initial configurations, respectively. Left: original peg-insert-side setup. Center: end-effector initially rotated 90 degrees about the vertical axis (corresponding to the left half of Figure 20). Right: peg initially rotated 90 degrees about the vertical axis (corresponding to the right half of Figure 20).

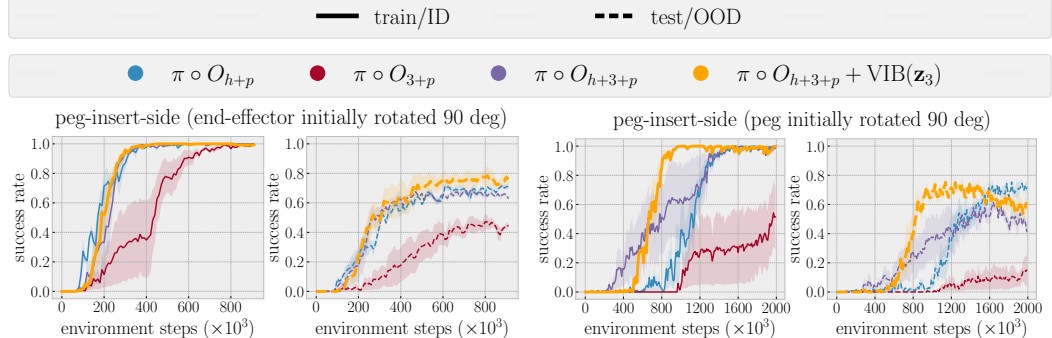

Figure 20: DrQ-v2 results for the two modified versions of the Meta-World peg-insert-side task (visualized in Figure 19). For the first modified version, we see generalization performance trends similar to those in the original rotation-less peg-insert-side task (second row, second column of Figure 6). In the second modified version, $\pi \circ O_{h+3+p} + \text{VIB}(\mathbf{z}_3)$ outperforms $\pi \circ O_{h+3+p}$ in terms of sample complexity and generalization, and $\pi \circ O_{h+p}$ in terms of sample complexity. However, test performance begins to droop after 1.6M steps – we attribute this to overfitting on the training distribution, which would likely occur to the other agents as well if they were trained post-convergence to a similar extent. Shaded regions indicate the standard error of the mean over three random seeds.

## C.7 HYPERPARAMETERS

We present the DrQ-v2 hyperparameters used in the Meta-World experiments in Table 10. The configuration is largely identical to the one used in the original DrQ-v2 algorithm (Yarats et al., 2021).

Table 10: The hyperparameters used for DrQ-v2 in the Meta-World tasks. The lower half of the table presents the hyperparameters that are specific to the regularization methods and ablation agents.

| parameter | setting |
|---|---|
| replay buffer capacity | 100,000 for {handle-press-side, button-press, reach-hard}; 400,000 for all other environments |
| action repeat | 2 |
| frame stack | 3 |
| seed frames | 4000 |
| exploration steps | 2000 |
| $n$-step returns | 3 |
| mini-batch size | 256 |
| discount $\gamma$ | 0.99 |
| optimizer | Adam |
| learning rate | $10^{-4}$ |
| agent update frequency | 2 |
| critic Q-function soft-update rate $\tau$ | 0.01 |
| features dim. | 50 |
| hidden dim. | 1024 |
| exploration stddev. clip | 0.3 |
| exploration stddev. schedule | $\mathrm{linear}(1.0, 0.1, 500000)$ |
| weight $\beta_3$ of KL div. term in $\pi \circ O_{h+3+p} + \mathrm{VIB}(\mathbf{z}_3)$ | 500 for {soccer, peg-insert-side-hard}; 50 for peg-insert-side w/ rotated initial gripper (for Appendix C.6) 1 for peg-insert-side w/ rotated initial peg (for Appendix C.6) 10 for all other environments |
| weights $\beta_h$, $\beta_3$ of KL div. terms in $\pi \circ O_{h+3+p} + \mathrm{VIB}(\mathbf{z}_h) + \mathrm{VIB}(\mathbf{z}_3)$ | $\beta_h$:   0.1 for {button-press, reach-hard, soccer};   0.001 for all other environments $\beta_3$:   same as $\beta_3$ of KL div. term in $\pi \circ O_{h+3+p} + \mathrm{VIB}(\mathbf{z}_3)$ |
| weight $\beta_3$ of KL div. term in $\pi \circ O_{3'+3+p} + \mathrm{VIB}(\mathbf{z}_3)$ | 0 for handle-press-side; 500 for soccer; 1 for all other environments |
| weight $\alpha_3$ of $\ell_2$ reg. term in $\pi \circ O_{h+3+p} + \ell_2(\mathbf{z}_3)$ | $10^8$ for all environments |

## D MISCELLANEOUS DETAILS

We applied an exponentially weighted moving average filter on the data for DrQ in Figure 2 ($\alpha = 0.6$), for DAC in Figure 3 ($\alpha = 0.3$), and for DrQ-v2 in Figures 6 and 17 ($\alpha = 0.5$) to smoothen the train and test curves for increased readability. The smoothing factor $\alpha$ lies in the range $[0, 1]$, where values closer to 0 correspond to more smoothing.

