# OpenReview forum: "Vision-Based Manipulators Need to Also See from Their Hands"
_ICLR.cc/2022/Conference — ICLR 2022 Oral_

### Official Review · Reviewer_z6Z4 · 2021-11-01

**Correctness:** 4
**Technical Novelty And Significance:** 3
**Empirical Novelty And Significance:** 3
**Recommendation:** 8
**Confidence:** 4

**Main Review:**

The paper is well motivated, clearly written and coherently structured. The exposition of the main ideas is linear and easy to follow. The contribution is clear, well presented and well motivated. The notation and the formulation of the proposed method are clearly presented. Claims are supported by thorough experimental results. Figures and tables are presented in a nice and easily readable way, and help grasping the contribution of the study. Videos are also helpful in understanding the experimental setup and the results in a qualitative way.
You mention tactile signals as a good example in robot manipulation of local streams of information. What is the applicability of your study to the tactile sensor modality?
It may be that some of the tasks could be solved using proprioceptive information only - do you have results using proprioception only as your observation space? Comparing the results presented in the paper with results obtained with proprioception only would be instrumental to understand the effect/contribution of vision (especially in the case of O_{h+p})


**Summary Of The Paper:**

This paper presents an interesting empirical evaluation of the role of visual perspective in learning and generalization in the context of physical manipulation. The paper compares performances using third-person and in-hand visual perspectives, showing that hand-centric vision consistently improves training and out-of-distribution generalization. Authors also explore a combination of the two perspectives by proposing to regularize the third-person information stream to maintain generalization performance.

**Summary Of The Review:**

The paper is well written, the contribution is clearly presented and the experimental results are thoroughly executed. An evaluation comparing the presented results with the proprioception-only case could further improve the analysis.

---

> ### Author Response · Authors · 2021-11-17
> **Author response to Reviewer z6Z4**
>
> Thank you for your thorough feedback. We appreciate that you took the time to comment on the aspects of the paper you liked. We have revised the paper with new discussion and experiments based on your feedback, and we respond to individual points below.
>
> > You mention tactile signals as a good example in robot manipulation of local streams of information. What is the applicability of your study to the tactile sensor modality?
>
> This is an excellent question. Our study is broadly related to the ideas of tactile/multi-modality sensing: we study the effects of sensor embodiment and how to properly fuse information from multiple sensors for the sake of generalization. We view both hand-centric cameras and tactile sensing as instances of hand-centric sensing. Since our work operates in just one modality (modulo proprioception), we are able to provide controlled insight on the effect of perspective that may apply to other forms of hand-centric sensing. For example, our work supports the idea that tactile sensing not only provides modality-specific advantages but also affords generalization through locality/symmetry. We have revised Section 6 of the paper to include more discussion on this.
>
> > It may be that some of the tasks could be solved using proprioceptive information only - do you have results using proprioception only as your observation space? Comparing the results presented in the paper with results obtained with proprioception only would be instrumental to understand the effect/contribution of vision (especially in the case of O_{h+p})
>
> Thank you for the insightful suggestion. We have run a proprioception-only variant of the DrQ-v2 agent on the Meta-World tasks, and have revised the paper to include a pointer to the results in Appendix B.5. Since the proprioception-only agent performs poorly during training for all of the tasks, this experiment concretely establishes that vision is necessary to consistently solve the tasks. Indeed, all of these tasks have wide distributions of initial object positions. Thus, a manipulator operating from proprioception only without privileged information is forced to do blind physical search to accomplish task instances.

---

> > ### Comment · Reviewer_z6Z4 · 2021-11-24
> > **score confirmed**
> >
> > Thank you for addressing my comments.

---

### Official Review · Reviewer_u4WJ · 2021-11-03

**Correctness:** 4
**Technical Novelty And Significance:** 2
**Empirical Novelty And Significance:** 2
**Recommendation:** 8
**Confidence:** 4

**Main Review:**

Specifics about the training
- 84x84 RGB input images.
- outputs 3D end-effector position relative to the robot base, 1D gripper width, and a boolean contact flag for each of two gripper fingers. No rotation.
- Three learning algorithms: Dagger, DrQ, DAC.

The experimental set up to show that hand perspective is better than third perspective involves picking up a cube (which is somewhat trivial task and naturally favours hand perspective). My first impression was that it's certainly helpful to have hand perspective as it always gets a close up / zoomed-in view of the object free of any occlusions etc. so it makes sense that the hand perspective performs better. However, if you have a more complicated manipulator e.g. hands the self occlusions from fingers and as the hand starts to cage the object, you'd need a third person view to get a better view of the object. Secondly, I feel adding rotation in the end-effector target (which the current output parameterisation doesn't include) could make things even for third perspective too. Is there a reason why the network only outputs just 3D end-effector position? For this particular task I believe you don't need rotation but any general task you'd need to also parameterise rotations as rotations will bring occlusions for hand perspective too.

In the 6 meta-world tasks, they show that combining a third perspective together with information bottleneck regularisation leads to better generalisation. Though most (if not all) of these tasks involve decision making that has more to do with the object they are reaching to and less about doing any long horizon interaction with scene or other objects in the scene after interaction with the object they are reaching to.

The paper seems to suggest that a zoomed-in view of the object by using a hand perspective almost always helps which I agree with. It seems to highlight the design choices that we regularly make when doing manipulation with raw observations need to be carefully looked into. Although this is not the first paper to show that e.g. https://arxiv.org/pdf/2012.07975.pdf have also shown that adding hand perspective helps. This paper was also not cited.




**Summary Of The Paper:**

The paper studies the effect of visual perspective by using the images coming from a camera installed on the hand of the robot in the specific context of robot manipulation from raw observations.  Results demonstrate that such a choice of visual perspective requires no algorithmic changes but can improve OOD generalisation and training efficiency. Of course this does not mean that we abandon altogether the traditional third person perspective as in many cases just having a close-up view of one object may not be enough when making decisions about the scene as a whole. Therefore, they also show that combined with third person perspective with information bottleneck regularisation can improve the OOD generalisation. They show results on six different manipulation tasks adapted from Meta-World and their choice of perspective improves OOD generalisation.

**Summary Of The Review:**

The paper is a good case study. The ideas presented in the paper are not something that's unknown but this paper does a good investigation on the choice of perspectives.

---

> ### Author Response · Authors · 2021-11-17
> **Author Response to Reviewer u4WJ**
>
> Thank you for your thoughtful review and constructive feedback. We have revised the paper with new discussion and experiments based on your feedback, and we respond to individual points below.
>
> >  Specifics about the training … The experimental set up to show that hand perspective is better than third perspective involves picking up a cube (which is somewhat trivial task and naturally favours hand perspective).
>
> This description and critique of the cube grasping task (Section 3) is fair, but this is only one part of our empirical study. The Meta-World results in Section 4 also contain head-to-head comparisons between the two perspectives ($\pi \circ O_{h+p}$ and $\pi \circ O_{3+p}$) for a variety of manipulation tasks, most of which are much less trivial than picking up a cube. Indeed, the “reach-hard” and “peg-insert-side-hard” tasks were specifically designed to be hard for the hand-centric perspective. As you correctly suggest, such tasks demonstrate that the hand-centric perspective alone is not sufficient in general, which we ultimately address via the $\pi \circ O_{h+3+p} + \mathrm{VIB}(z_3)$ agent.
>
> > My first impression was that it's certainly helpful to have hand perspective as it always gets a close up / zoomed-in view of the object free of any occlusions etc. so it makes sense that the hand perspective performs better. However, if you have a more complicated manipulator e.g. hands the self occlusions from fingers and as the hand starts to cage the object, you'd need a third person view to get a better view of the object.
>
> Dextrous manipulation from vision is typically done palm-up to avoid occlusions. Tactile sensing becomes necessary otherwise; we don’t think that vision is in general sufficient for manipulation.  The relevance of our work to tactile sensing is that it supports the idea that tactile sensing not only provides modality-specific advantages (e.g. no issue with occlusions) but also affords generalization through locality/symmetry.
>
> > I feel adding rotation in the end-effector target (which the current output parameterisation doesn't include) could make things even for third perspective too. Is there a reason why the network only outputs just 3D end-effector position? For this particular task I believe you don't need rotation but any general task you'd need to also parameterise rotations as rotations will bring occlusions for hand perspective too.
>
> The reason is as the reviewer suggests -- all the tasks we considered are solvable without changes in orientation. To probe this, we have added Appendix B.6 to the paper, which features additional experiments on two variants of peg-insert-side that require an additional 1-DoF end-effector orientation control. On both variants, the hand-centric perspective still affords faster training and better generalization than the third-person perspective, and both perspectives with VIB regularization dominates. We leave experiments with 6-DoF pose control to future work.
>
> > Though most (if not all) of these tasks involve decision making that has more to do with the object they are reaching to and less about doing any long horizon interaction with scene or other objects in the scene after interaction with the object they are reaching to.
>
> Thank you for pointing this out. It is fair to be skeptical of how applicable our study is to long-horizon tasks and scene-level reasoning. However, it should be noted that long-horizon manipulation tasks will involve sub-tasks that are analogous to those we experiment with, so the insights from our study should at least apply in that respect. We have added this discussion to Section 6 of our paper.
>
> > Although this is not the first paper to show that e.g. https://arxiv.org/pdf/2012.07975.pdf have also shown that adding hand perspective helps. This paper was also not cited.
>
> Thank you for the relevant reference. We have added it to the first paragraph of the related works section. Like the other works we cited (e.g. Wu et al. (2021), Mandelkar et al. (2021)), Zhan et al. (2020) do not consider out-of-distribution generalization, which is a large focus of our empirical study.

---

> ### Author Response · Authors · 2021-11-23
> **Re: Your Concern About Novelty**
>
> Dear Reviewer u4WJ,
>
> We notice that your main concern regarding our work is novelty. You mention in your summary that “The ideas presented in the paper are not something that’s unknown”, and you give scores of 2 for “Novelty And Significance”. It seems that you are largely satisfied with significance, as you deem the work “a good case study … a good investigation”. We agree that our ideas are not completely novel — people have put cameras in robot hands. We have better clarified the novelty of our work by adding the following sentence to the abstract:
>
> > While some practitioners have long put cameras in the hands of robots, our work systematically analyzes the benefits of doing so and provides simple and broadly applicable insights for improving vision-based robotic manipulation.
>
> We hope this improves your opinion of our work. Thank you again for your reviewing efforts.

---

> > ### Comment · Reviewer_u4WJ · 2021-11-24
> > **final remarks**
> >
> > Hi there,
> >
> > Thank you for your rebuttal. It more than answers my concerns and thanks for your explanations. I am very supportive of this work and as I said before it is a useful case study to understand the design choices we make when doing any image based robot control. Thank you also for Appendix B.6.

---

> > > ### Comment · Reviewer_u4WJ · 2021-11-24
> > > **improved the review score**
> > >
> > > The score now reads 8 which I forgot to mention in my previous message --- sorry about that.

---

> > > > ### Author Response · Authors · 2021-12-01
> > > > **Acknowledgement**
> > > >
> > > > Thank you very much for your engagement.

---

### Official Review · Reviewer_RmQW · 2021-11-04

**Correctness:** 4
**Technical Novelty And Significance:** 4
**Empirical Novelty And Significance:** 4
**Recommendation:** 8
**Confidence:** 3

**Main Review:**

### Strengths
- Well motivated idea
- Compelling results
- Overall clearly written
- Good ablations and the idea is shown for multiple algorithms and in a wide variety of settings
- Interesting approach to generalize the hand camera to settings with a higher degree of partial observability

### Weakness

- $$z_\text{shift}$$ is only explained in the appendix, it should be explained in the main paper.
- The information bottleneck technique harms performance initially



### Suggestions for improvement

- In concurrent work, Szot et al (NuerIPS 2021) also used a hand/arm camera to learn manipulation policies and found similar trends. It may be worth citing as additional support for these finding.

- How useful is the data-aug in DrQ for the hand camera? Part of the argument for DrQ is to reduce overfitting of the Q function during training. Perhaps with the hand camera you no longer need aug?

- What about recurrent policies instead of the third person camera?


Szot et al: https://arxiv.org/abs/2106.14405

**Summary Of The Paper:**

This paper presents an analysis of camera placement for vision-based manipulators. Specifically it compares the performance of a disembodied third person camera vs. place the camera on the robot's hand/gripper.

The authors find that the hand camera improves generalization and training performance in the cases where a hand camera still reveals enough information to complete the task.

When the hand camera does not reveal enough information to perform the task, the third person camera is still needed and the authors propose to use an information bottleneck to reduce the amount of information used from the third person camera, thereby improving generalization even when it's needed.


**Summary Of The Review:**

This paper presents thorough analysis of using a 3rd person camera vs. a hand camera and finds that hand cameras generalize better. I believe this work presents a useful contribution.


### Post Rebuttal Update

I thank the authors for their response. I continue to think this is a good paper that should be accepted for publication.

---

> ### Author Response · Authors · 2021-11-17
> **Author Response to Reviewer RmQW**
>
> Thank you for your precise review and constructive feedback. We have revised the paper with new discussion and experiments based on your feedback, and we respond to individual points below.
>
> ### Weaknesses
> > $z_\mathrm{shift}$ is only explained in the appendix, it should be explained in the main paper.
>
> Thank you for pointing this out. Our updated draft explains this in the caption of Figure 2.
>
> > The information bottleneck technique harms performance initially.
>
> Indeed, incorporating the VIB does slow training (but does not lower asymptotic performance) for some tasks. In the discussion on p. 8 (under the heading “Effect of regularizing the third-person information stream via a VIB.”), we conjecture based on work studying how neural networks learn spurious correlations that this slowdown may be hard to avoid, at least in the single-task setting. In future work, it would be interesting to explore ways to combat this, e.g. via an inductive bias, side information, multi-task learning, etc.
>
> ### Suggestions for improvement
>
> > In concurrent work, Szot et al. (NuerIPS 2021) also used a hand/arm camera to learn manipulation policies and found similar trends. It may be worth citing as additional support for these finding.
>
> Thank you for the relevant reference. We have added discussion on Szot et al. to the second paragraph of our related works section. It is encouraging to see complementary results on the significant effect of sensor embodiment on generalization. One key difference between our setting and theirs is that we consider physical manipulation tasks, whereas their picking task uses “abstracted grasping” and hence asks the policy to focus on learning to navigate and move the manipulator in free space.
>
> > How useful is the data-aug in DrQ for the hand camera? Part of the argument for DrQ is to reduce overfitting of the Q function during training. Perhaps with the hand camera you no longer need aug?
>
> This is an intriguing suggestion. We experiment with removing the data augmentation in DrQ (essentially recovering SAC) for both $\pi \circ O_h$ and $\pi \circ O_3$ and have revised the paper to point to new results in Appendix A.5. The results show that data augmentation remains important even with the hand-centric perspective: removing the data augmentation severely compromises training for both agents, though $\pi \circ O_h$ still performs better.
>
> > What about recurrent policies instead of the third person camera?
>
> This is a good suggestion: memory could certainly be used in place of a third-person camera as a strategy to resolve partial observability. However, recurrent networks are known to be difficult to train in an RL setting, and their use in the literature seems to correspond to when observations that would resolve the partial observability cannot be conveniently instrumented. Since this is not the case in tabletop manipulation, we follow prior work in choosing to use third-person observations with feedforward policies. In contrast, recurrence is standard practice for tasks involving navigation as global third-person observations are less natural: Hill et al. (2019) and Szot et al. (2021) follow this paradigm. In our updated draft, we have added this discussion to the second paragraph of our related works section. It would be interesting to study the performance trade-offs between the two strategies in future work.

---

### Author Response · Authors · 2021-11-20
**Summary of Revisions**

We thank the reviewers again for their thoughtful reviews and their appreciation of the paper's contributions, experimental execution, and writing. We were eager to incorporate their constructive feedback as revisions to the paper (highlighted in magenta in the pdf), which we now summarize:

### Experiments
- [RmQW] In Appendix A.5, we study whether the data augmentation in DrQ is still necessary with the hand-centric perspective. The results indicate that this is indeed the case.
- [u4WJ] In Appendix B.6, we study whether orientation control could change the relative performance between choices of perspective. We find that the three agents that used a hand-centric perspective significantly outperformed the one that didn't. We also find that VIB regularization results in improvement when using both perspectives. These results are in line with what we observed in the previous experiments.
- [z6Z4] In Appendix B.5, we study to what extent the Meta-World tasks we consider require vision by running a proprioception-only agent. This agent fails to consistently solve the tasks on the training distribution, demonstrating the importance of vision.

### Writing
- [RmQW] We add discussion on the choice of memory-augmented policies vs. global observations to resolve partial observability to our related works section.
- [u4WJ] We add discussion on the applicability of our study to more complex, longer-horizon manipulation tasks in the conclusion.
- [z6Z4] We make explicit the possible relevance of our findings on the role of perspective in manipulation to robotic systems that use tactile sensing in the conclusion.
- [RmQW] We add an explanation of $z_\text{shift}$ in the main paper (caption of Figure 2).

### Additional References
- [RmQW] We add discussion of Szot et al. (2021) to our related works section.
- [u4WJ] We add Zhan et al. (2020) to our related works section.

We hope that the additional experimental results and text have addressed the reviewers' concerns, and we are grateful for their help in improving the paper. Please let us know if you have any further comments!

---

> ### Author Response · Authors · 2021-11-23
> **Additional Revision**
>
> ### Writing
> - [u4WJ] We add clarifications on the novelty of our work to the abstract (currently visible in the pdf but not on this webpage).

---

### Decision · Program_Chairs · 2022-01-20

**Decision:**

Accept (Oral)

**Comment:**

All reviewers consistently agree on the high quality of the research presented in this paper, such that it the paper clearly is significantly above the acceptance threshold of ICLR.